# DEGREE-CONSCIOUS SPIKING GRAPH FOR CROSS-DOMAIN ADAPTATION

## ABSTRACT

Spiking Graph Networks (SGNs) have demonstrated significant potential in graph classification by emulating brain-inspired neural dynamics to achieve energy-efficient computation. However, existing SGNs are generally constrained to in-distribution scenarios and struggle with distribution shifts. In this paper, we first propose the domain adaptation problem in SGNs, and introduce a novel framework named **De**gree-Conscious **S**piking **Gra**ph for Cross-**D**omain **A**daptation (DeSGraDA). DeSGraDA enhances generalization across domains with three key components. First, we introduce the degree-conscious spiking representation module by adapting spike thresholds based on node degrees, enabling more expressive and structure-aware signal encoding. Then, we perform temporal distribution alignment by adversarially matching membrane potentials between domains, ensuring effective performance under domain shift while preserving energy efficiency. Additionally, we extract consistent predictions across two spaces to create reliable pseudo-labels, effectively leveraging unlabeled data to enhance graph classification performance. Furthermore, we establish the first generalization bound for SGDA, providing theoretical insights into its adaptation performance. Extensive experiments on benchmark datasets validate that DeSGraDA consistently outperforms state-of-the-art methods in both classification accuracy and energy efficiency.

## 1 INTRODUCTION

Spiking Graph Networks (SGNs) (Zhu et al., 2022; Xu et al., 2021b) as a specialized neural network combining Spiking Neural Networks (SNNs) (Gerstner & Kistler, 2002; Maass, 1997) with Graph Neural Networks (GNNs) (Kipf & Welling, 2017; Scarselli et al., 2009), have emerged as a ground-breaking paradigm in artificial neural networks, uniquely designed to process graph-structured data by mimicking the bio-inspired mechanisms of the human brain. SGNs convert graph features into binary spiking signals, replacing matrix multiplications with simple additions to achieve high energy efficiency. They further exploit temporal spiking representations, encoding information in spike timing to enable asynchronous, event-driven processing. It is particularly critical for applications where energy consumption is a bottleneck, such as real-time brain-computer interfaces (Kumar et al., 2022; Nason et al., 2020), large-scale sensor networks (Yao et al., 2021; Wilson et al., 2024), and temporal analysis (Yin et al., 2024; Zhu et al., 2024; Zhou et al., 2021).

Despite their potential for energy-efficient graph representation, existing SGNs are primarily studied under closed-world assumptions, where source and target data share identical distributions (Yin et al., 2024; Duan et al., 2024). This assumption is inadequate for many real-world scenarios, such as brain–computer interfaces (BCIs) (Binnie & Prior, 1994; Biasiucci et al., 2019) where distribution shifts degrade performance (Zhao et al., 2020; 2021). Although recent advancements in transfer learning for SNNs have shown promise in vision tasks by leveraging neuromorphic adaptations (Zhan et al., 2024; Guo et al., 2024), they are primarily designed for grid-like inputs and fail to generalize to graph data. The non-Euclidean nature and inherent irregularity of graphs introduce fundamental challenges (Bronstein et al., 2017), as existing methods neglect topological dependencies and message-passing mechanisms crucial for effective graph learning. Consequently, directly applying these methods to SGNs results in suboptimal adaptation under domain shifts.

In this paper, we investigate the development of energy-efficient SGNs for scenarios involving distribution shifts. However, designing an effective domain adaptation framework for SGNs poses

several fundamental challenges: (1) *How to adapt spiking representations to account for the structural diversity of graph-structured data?* Traditional SGNs typically assign a fixed firing threshold to all nodes, ignoring the structural diversity in graphs (Xu et al., 2021a; Yin et al., 2024). This uniform treatment leads to under-activation in nodes with fewer connections, where important features are missed, and over-activation in highly connected nodes, where excessive signals distort the learned representation. Both cases reduce the model's representational effectiveness. (2) *How to design domain adaptation strategies that account for the temporal-based representations?* Unlike static neural models, SGNs encode information through temporal spike sequences, making them more sensitive to domain shifts (Zhou et al., 2023; Zhan et al., 2021). Existing methods fail to address how these shifts impact the spike sequences, resulting in suboptimal alignment across domains. (3) *How to theoretically characterize and bound the generalization error of SGNs under domain shift?* Despite empirical advances (Zhang et al., 2021; Zhan et al., 2024), the theoretical understanding of SGN domain adaptation remains limited. Without a principled framework to quantify generalization under distribution shift, it is difficult to design adaptation methods with guaranteed performance.

To tackle these challenges, we propose a framework called **De**gree-Conscious **S**piking **Gra**ph for Cross-**D**omain **A**daptation (DeSGraDA). This has three components: (1) degree-conscious spiking representation, which assigns variable firing thresholds to nodes based on their degrees, enabling adaptive control over spiking sensitivity. This degree-conscious mechanism balances the firing frequencies of high- and low-degree nodes, preventing information loss in sparsely connected nodes and avoiding distortion from excessive signals in highly connected ones, thus enhancing the expressiveness of the spiking representations; (2) temporal distribution alignment, which explicitly aligns the time-evolving spiking representations between the source and target domains. By leveraging membrane potential dynamics as evolving signals, the model captures domain-specific patterns, improving robustness to temporal shifts; and (3) pseudo-label distillation assigns reliable pseudo-labels by aligning consistent predictions from shallow and deep network layers. We also demonstrate that this pseudo-label distillation module can effectively reduce a generalization bound tailored for spiking graph domain adaptation. In summary, DeSGraDA provides a simple yet effective solution to a novel and underexplored problem, and offers deep insights into the spiking graph domain adaptation.

Our contributions can be summarized as follows: (1) **Problem Formulation:** We first introduce the spiking graph domain adaptation for graph classification, highlighting the challenges posed by the inflexible threshold mechanism of SGNs and theoretical limitations that hinder effective adaptation. (2) **Novel Architecture and Theoretical Analysis:** We propose DeSGraDA, a framework combining degree-conscious spiking representation and temporal distribution alignment. Moreover, we provide a generalization bound for spiking graph domain adaptation. (3) **Extensive Experiments**. We evaluate the proposed DeSGraDA on extensive spiking graph domain adaptation learning datasets, demonstrating that it can outperform various state-of-the-art methods.

## 2 RELATED WORK

**Domain Adaptation (DA).** DA transfers knowledge from a labeled source domain to an unlabeled target domain by mitigating the distributional shift between the two domains (Redko et al., 2017; Long et al., 2018; Shen et al., 2018). It has been widely applied to vision and language tasks (Wei et al., 2021b;a; Shi et al., 2024). Recently, DA has been extended to graph data to address the unique challenges posed by complex relationships, leading to the emergence of Graph Domain Adaptation (GDA) (You et al., 2023; Liu et al., 2024a; Cai et al., 2024). Most existing GDA approaches first leverage GNNs to generate node and graph representations (Zhu et al., 2021; Yin et al., 2022; Liu et al., 2024b), followed by adversarial learning to implicitly align feature distributions and reduce domain discrepancies. They also apply structure-aware strategies to explicitly align graph-level semantics and topological structures, incorporate spectral contrastive alignment to capture domain-invariant spectral patterns, and enforce smoothness constraints to promote consistent feature propagation across domains (Luo et al., 2024; Zhang et al., 2025; Chen et al., 2025b). However, GDA remains underexplored in the context of spiking graphs, where energy efficiency becomes a critical requirement for real-world applications. To bridge this gap, we introduce the novel problem of Spiking Graph Domain Adaptation (SGDA), extending GDA to spiking graphs for energy-efficient domain adaptation.

**Spiking Graph Networks (SGNs).** SGNs are a specialized neural network combining SNNs (Gerstner & Kistler, 2002; Maass, 1997) with GNNs, preserving energy efficiency while achieving competitive performance on various tasks (Li et al., 2023; Yao et al., 2023; Duan et al., 2024).

Existing research on SGNs focuses on capturing temporal information within graphs and enhancing scalability. For instance, (Xu et al., 2021a) utilizes spatial-temporal feature normalization within SNNs to effectively process dynamic graph data, ensuring robust learning and improved performance. (Zhao et al., 2024) proposes a method that adapts to evolving graph structures through a novel architecture that updates node representations in real time. Additionally, (Yin et al., 2024) adapts SNNs to dynamic graph settings and employs implicit differentiation for the node classification task. However, existing methods still suffer from data distribution shift issues when the training and test data come from different domains, resulting in degraded performance and generalization. To tackle these challenges, we propose a novel domain adaptation method for spiking graph networks.

## 3 PRELIMINARIES

**Problem Setup.** Given a graph $G = (V, E, \mathbf{X})$ with node set $V$, edge set $E$, and node attribute matrix $\mathbf{X}$. To construct spiking-compatible inputs for SGNs, we sample binary features $S$ from the Bernoulli distribution with probability of $\mathbf{X}$ (i.e., $S \sim Ber(\mathbf{X})$) as input of SGNs (Zhao et al., 2024). In this paper, we focus on the problem of spiking GDA for graph classification. The source domain $\mathcal{D}^s = \{(G_i^s, y_i^s)\}_{i=1}^{N_s}$ is labeled, where $N^s$ is the number of source-domain graphs $G_i^s$ and $y_i^s$ is the label of $G_i^s$. The target domain $\mathcal{D}^t = \{G_j^t\}_{j=1}^{N_t}$ is unlabeled and contains $N^t$ graphs. Both domains share the same label space $\mathcal{Y}$ but can have different graph topologies or attribute distributions.

**Domain Adaptation with Optimal Transport (OT).** Following (You et al., 2023), we factorize a trained model $h$ as $g \circ f$, where $f : \mathcal{D} \mapsto \mathbb{R}^d$ is the feature extractor ($Z = f(\mathcal{D})$) and $g : \mathbb{R}^d \mapsto \mathcal{Y}$ is the discriminator ($Y = g(Z)$). For simplicity, we focus on binary classification with $\mathcal{Y} = [0, 1]$. Denote the classifier predicting labels from the feature representation as $\hat{g} : \mathbb{R}^d \mapsto \mathcal{Y}$. The source and target risks are given by $\hat{\epsilon}_S(g, \hat{g}) = \frac{1}{N_s} \sum_{n=1}^{N_s} |g(Z_n) - \hat{g}(Z_n)|$ and $\epsilon_T(g, \hat{g}) = \mathbb{E}_{\mathbb{P}_T(Z)} |g(Z) - \hat{g}(Z)|$. DA with OT (Villani et al., 2008) addresses covariate shift by optimally transporting masses between source and target distributions while minimizing cost. Theorem 1 shows that the generalization gap depends on both domain divergence $2C_g W_1(\mathbb{P}_S(Z), \mathbb{P}_T(Z))$ and model discriminability $\omega$.

**Theorem 1** *(Shen et al., 2018) Assume that the discriminator $g$ is $C_g$-Lipschitz. Let $\mathcal{H} := \{g : \mathcal{Z} \to \mathcal{Y}\}$ (where $\mathcal{Z}$ is the feature space) be the set of bounded real-valued functions with pseudo-dimension $Pdim(\mathcal{H}) = d$. For any $g \in \mathcal{H}$, the following holds with probability at least $1 - \delta$:*

$$\epsilon_T(g, \hat{g}) \leq \hat{\epsilon}_S(g, \hat{g}) + \sqrt{\frac{4d}{N_S} \log\left(\frac{eN_S}{d}\right) + \frac{1}{N_S} \log\left(\frac{1}{\delta}\right)} + 2C_g W_1(\mathbb{P}_S(Z), \mathbb{P}_T(Z)) + \omega,$$

*where $\omega = \min_{\|g\|_{Lip} \leq C_g} \{\epsilon_S(g, \hat{g}) + \epsilon_T(g, \hat{g})\}$ is the discriminative ability to capture source and target data, and $W_1(\mathbb{P}, \mathbb{Q})$ is the distribution divergence defined in (Redko et al., 2017; Shen et al., 2018).*

Applying OT-based DA methods to graph data introduces challenges due to the non-Euclidean nature of graphs and intricate dependencies between nodes. You et al. (2023) extends the OT-based DA framework to graphs and provides a generalization bound for GDA. Details are in Appendix A.

**Spiking Graph Networks (SGNs).** Spiking Neural Networks (SNNs)(Maass, 1997; Gerstner & Kistler, 2002; Bohte et al., 2000) are brain-inspired models offering notable advantages in temporal information processing and energy efficiency. More details are in Appendix B. SGNs are a specialized form of SNNs tailored for graph data (Xu et al., 2021a; Zhu et al., 2022), where each node is modeled as a spiking neuron. The membrane of each node evolves based on both the temporal spiking dynamics and the graph's structural connectivity. Let $u_{\tau,i}$ be the membrane potential of node $i$ at latency step $\tau$. SGNs first updates the membrane potentials via input aggregation:

$$u_{\tau+1,i} = \lambda(u_{\tau,i} - V_{th} s_{\tau,i}) + \sum_j w_{ij} \mathcal{A}(A, s_{\tau,j}) + b_i, \tag{1}$$

where $s_{\tau,i}$ is the spiking representation, $\lambda \in (0, 1)$ is the leak factor, $V_{th}$ is the firing threshold, $b_i$ is the bias, $w_{ij}$ is the synaptic weight from node $j$ to node $i$, $A$ is the adjacency matrix, and $\mathcal{A}$ is the graph-based aggregation operation. Next, spikes are triggered through thresholding:

$$s_{\tau+1,i} = \mathbb{H}(u_{\tau+1,i} - V_{th}), \quad u_{\tau+1,i} = (1 - s_{\tau+1,i})u_{\tau+1,i} + s_{\tau+1,i}V_{reset}, \tag{2}$$

where $V_{\text{reset}}$ is the reset potential, and $\mathbb{H}(x)$ is the Heaviside step function, which serves as the non-differentiable spiking function. Finally, the neuron is reset upon firing.

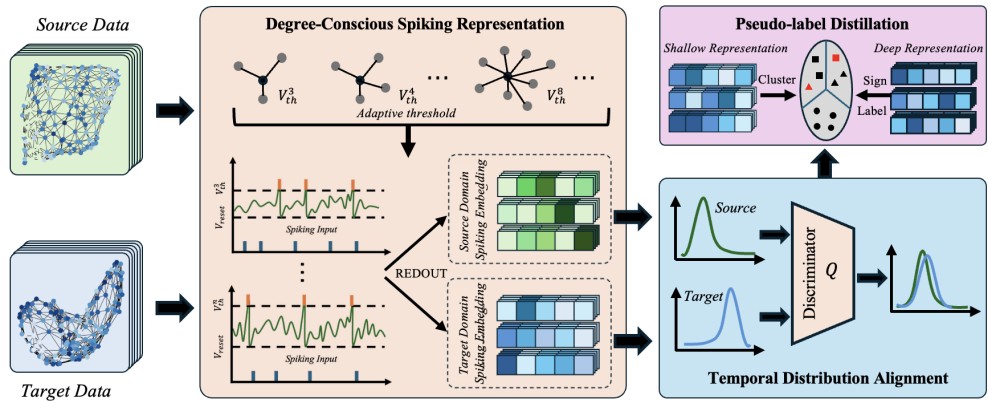

Figure 1: Overview of the proposed DeSGraDA. Degree-Conscious Spiking Representation generates source and target domain spiking representations by adapting neuron firing threshold based on node degrees. To align domain distributions, Temporal Distribution Alignment leverages adversarial training on temporal membrane potential to counter domain discrimination. Furthermore, Pseudo-label Distillation is employed to identify reliable target samples and reinforce overall model performance.

# 4 PROPOSED METHODOLOGY

This paper proposes a novel framework DeSGraDA for the problem of spiking graph domain adaptation. DeSGraDA consists of three parts: **Degree-Conscious Spiking Representation** (Section 4.1), which assigns adaptive firing thresholds based on node degrees to address the limitations of rigid, fixed-threshold architectures; **Temporal Distribution Alignment** (Section 4.2), which employs adversarial training on temporal membrane potentials against a domain discriminator to align spiking dynamics across domains; and **Pseudo-label Distillation** (Section 4.3) further applies the pseudo-label to enhance model performance. We also provide a theoretical generalization bound to support the effectiveness of DeSGraDA. An overview of the framework is in Figure 1.

## 4.1 DEGREE-CONSCIOUS SPIKING REPRESENTATION

First, we study the disadvantages of directly applying SNNs to graphs and then propose the degree-conscious spiking representation module. Existing SGNs (Li et al., 2023; Duan et al., 2024) usually employ a fixed global threshold ($V_{th}$ in Eq. 1 and 2) for firing. Assume that the membrane potential of node $u_{\tau,i}$ follows the normal distribution $\mathcal{N}(\mu, \sigma^2)$ (Kipf & Welling, 2016).

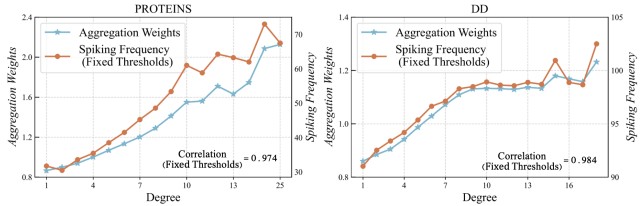

Figure 2: Correlation between spiking frequency and aggregation weights under fixed thresholds on different datasets.

The following Proposition shows that the high-degree nodes are more likely to trigger spikes than the low-degree ones. Proof is in Appendix C.

**Proposition 1** *With aggregation operation in SGNs (i.e., $\mathcal{A}$ in Eq. 1), the expectation of the updated node membrane potential is:* $\mathbb{E}(u_{\tau+1,i}) \sim \mathcal{N}\left(\left(1 + \sum_{j \in N(i)} w_{ij}\right)\mu, \left(1 + \sum_{j \in N(i)} w_{ij}\right)^2 \sigma^2\right),$ *where $N(i)$ is the set of node $i$'s neighbors, and $w_{ij}$ is the weight between nodes $i$ and $j$.*

From Proposition 1, we observe that node $i$ follows a normal distribution with a mean of $(1 + \sum_{j \in N(i)} w_{ij})\mu$, determined by the aggregated weights of its neighboring nodes We conduct an experiment to investigate the relationship between the aggregated weight ($\sum_{j \in N(i)} w_{ij}$) and spiking frequency (i.e., count of $u_T > V_{th}$) for a fixed threshold $V_{th}$. As shown in Figure 2, the spiking frequency and aggregation weights under fixed thresholds exhibit a relatively high correlation coefficient, indicating that nodes with higher degrees tend to accumulate more features from neighbors, making it easier to trigger spikes than those with fewer neighbors. Fixed thresholds inherently bias

spiking activation toward high-degree nodes, leading to under-representation of structurally important yet sparsely connected nodes and undermining both expressiveness and generalization in SGNs.

To alleviate this problem, we propose the use of degree-conscious thresholds. Specifically, let $B_i^s$ be the set of degrees for the nodes in graph $G_i^s$. We collect all unique node degrees across the source domain graphs with $B^s = set(B_1^s \cup \cdots \cup B_{N_s}^s)$, where $set(\cdot)$ operation is an unordered sequence of distinct elements. With the observation from Proposition 1, we propose setting higher thresholds for high-degree nodes and lower for low-degree nodes. For node $v$ with degree $d_v^s$, we have:

$$s_{\tau,v}^{d_v^s} = \mathbb{H}(u_{\tau,v} - V_{th}^{d_v^s}), \quad V_{th,\tau+1}^{d_v^s} = (1-\alpha)V_{th,\tau}^{d_v^s} + \alpha \bar{s}_{\tau,v}^{d_v^s},$$

$$u_{\tau+1,v} = \lambda(u_{\tau,v} - V_{th}s_{\tau,v}^{d_v^s}) + \sum\nolimits_{j \in N(v)} w_{ij}\mathcal{A}\left(A, s_{\tau,v}^{d_j^s}\right) + b_i, \tag{3}$$

where $V_{th}^{d_v^s}$ is the threshold for nodes with degree $d_v^s \in B^s$, initially set to $V_{th}$, and $\alpha$ is a hyper-parameter. $\bar{s}_\tau^{d_v^s}$ is the average of spiking representation $s^{d_v^s}$ with degree $d_v^s$ on latency step $\tau$. In Eq. 3, we dynamically update the threshold $V_{th,\tau+1}^{d_v^s}$ with $(1-\alpha)V_{th,\tau}^{d_v^s} + \alpha \bar{s}_\tau^{d_v^s}$ based on the average spiking frequency $\bar{s}_{\tau,v}^{d_v^s}$ of nodes with degree $d_v^s$. Consequently, high-degree nodes typically exhibit higher $\bar{s}_{\tau,v}^{d_v^s}$, resulting in increased thresholds $V_{th,\tau+1}^{d_v^s}$ to suppress over-activation, while low-degree nodes yield lower thresholds to enhance activation, enabling adaptive spiking control across structurally diverse nodes. For each node $v$ in graph $G_i^s$, we calculate the membrane potential $u_{\tau,v}^s$, and summarize all the node representations with a readout function (Xu et al., 2018) into the graph-level representation:

$$\mathbf{s}_{G_i^s} = \text{READOUT}\left(\{s_T^{d_v^s}\}_{v \in G_i^s}\right), \tag{4}$$

where $T$ is the total number of latency steps. Finally, we output the prediction $\hat{y}_{G_i^s}^s = H(\mathbf{s}_{G_i^s})$ with a classifier $H(\cdot)$ by minimizing the source classification loss $\mathcal{L}_S = \mathbb{E}_{G_i^s \in \mathcal{D}^s} l(y_{G_i^s}^s, \hat{y}_{G_i^s}^s)$, where $l(\cdot)$ is the loss function and $y_i^s$ is the ground truth of graph $G_i^s$ in the source domain.

### 4.2 TEMPORAL DISTRIBUTION ALIGNMENT

Unlike GNNs, spiking models rely on membrane potential dynamics to generate sparse spike trains, making their spike representations highly dynamic and non-differentiable. Existing GDA (Yin et al., 2023; Chen et al., 2025a) methods assume continuous, static embeddings and thus fail to align the time-dependent neural dynamics in SGNs. To address this, we propose a temporal-based alignment framework that captures and matches the evolution of spiking patterns across domains.

First, we introduce a temporal attention mechanism that adaptively aggregates time-dependent neuronal states. Specifically, given the temporal membrane potential of graphs on each latency step, we have $[\mathbf{u}_{1,G_i}, \ldots, \mathbf{u}_{\tau,G_i}, \cdots, \mathbf{u}_{T,G_i}]$, where $\mathbf{u}_{\tau,G_i} = \text{READOUT}(\{u_{\tau,v}\}_{v \in G_i})$ and $G_i \in \{\mathcal{D}^s, \mathcal{D}^t\}$. We stack the $T$ steps membrane potential into $\mathbf{U}_{G_i} \in \mathbb{R}^{T \times d}$, where $d$ is the hidden dimension. The goal is to learn an importance-weighted $\alpha_\tau$ to summarize of temporal membrane potential representation, which is formulated as follows:

$$\alpha_\tau = \text{Atten}(\mathbf{U}_{G_i}), \quad \tilde{\mathbf{U}}_{G_i} = \sum\nolimits_{\tau=1}^{T} \alpha_\tau \mathbf{u}_{\tau,G_i}, \tag{5}$$

where $\text{Atten}(\cdot)$ is the self-attention operation (Vaswani et al., 2017). Then, we propose the adversarial distribution alignment module to eliminate the discrepancy between the source and target domains. Specifically, for each source graph $G_i^s$ and target graph $G_j^t$, we denote the semantic classifier as $H(\cdot)$ to produce predicted labels, and a domain discriminator $Q(\cdot)$ to distinguish features from the source and target domains. The temporal-based distribution alignment module is adversarially trained to align the feature spaces of the source and target domains, which is formulated as:

$$\mathcal{L}_{AD} = \mathbb{E}_{G_i^s \in \mathcal{D}^s} \log Q\left(H(\tilde{\mathbf{U}}_{G_i}^s)|V_{th}^{B^s}\right) + \mathbb{E}_{G_j^t \in \mathcal{D}^t, \exists d_j^t \notin B^s} \log\left(1 - Q\left(H(\tilde{\mathbf{U}}_{G_i}^t)|V_{th}^{B^s}, V_{th}^{B^t}\right)\right),$$

where $B^t = \{d_i^t | d_i^t \in B^t, d_i^t \notin B^s\}$. We iteratively update $V_{th}^{B^t}$ with Eq. 3 on each latency. Furthermore, we present an upper bound for temporal-based distribution alignment.

**Theorem 2** *Assume that the learned discriminator is $C_g$-Lipschitz continuous, the feature extractor $f$ is $C_f$-Lipschitz that $||f||_{Lip} = \max_{G_1,G_2} \frac{||f(G_1)-f(G_2)||_2}{\eta(G_1,G_2)} = C_f$ for some graph distance measure*

*$\eta$ and the loss function bounded by $C > 0$. Let $\mathcal{H} := \{h : \mathcal{G} \to \mathcal{Y}\}$ be the set of bounded functions with the pseudo-dimension $Pdim(\mathcal{H}) = d$ that $h = g \circ f \in \mathcal{H}$, and provided the spike training data set $S_n = \{(\mathbf{X}_i, y_i) \in \mathcal{X} \times \mathcal{Y}\}_{i \in [n]}$ drawn from $\mathcal{D}^s$, with probability at least $1 - \delta$ the inequality :*

$$\epsilon_T(h, \hat{h}_T(\mathbf{X})) \leq \hat{\epsilon}_S(h, \hat{h}_S(\mathbf{S})) + 2\mathbb{E}\left[\sup \frac{1}{N_S}\sum_{i=1}^{N_S} \epsilon_i h(\mathbf{X}_i, y_i, p_i)\right] + C\sqrt{\frac{ln(2/\delta)}{N_S}} \tag{6}$$
$$+ \omega + 2C_f C_g W_1\left(\mathbb{P}_S(G), \mathbb{P}_T(G)\right),$$

*where the (empirical) source and target risks are $\hat{\epsilon}_S(h, \hat{h}(\mathbf{S})) = \frac{1}{N_S}\sum_{n=1}^{N_S} |h(\mathbf{S}_n) - \hat{h}(\mathbf{S}_n)|$ and $\epsilon_T(h, \hat{h}(\mathbf{X})) = \mathbb{E}_{\mathbb{P}_T(G}\{|h(G) - \hat{h}(G)|\}$, respectively, where $\hat{h} : \mathcal{G} \to \mathcal{Y}$ is the labeling function for graphs and $\omega = \min\left(|\epsilon_S(h, \hat{h}_S(\mathbf{X})) - \epsilon_S(h, \hat{h}_T(\mathbf{X}))|, |\epsilon_T(h, \hat{h}_S(\mathbf{X})) - \epsilon_T(h, \hat{h}_T(\mathbf{X}))|\right)$, $\epsilon_i$ is the Rademacher variable and $p_i$ is the $i^{th}$ row of $\mathbf{P}$, which is the probability matrix with:*

$$\mathbf{P}_{kt} = \begin{cases} \exp\left(\frac{u_k(t) - V_{th}}{\sigma(u_k(t) - u_{reset})}\right), & if \quad u_\theta \leq u(t) \leq V_{th}, \\ 0, & if \quad u_{reset} \leq u_k(t) \leq u_\theta. \end{cases}$$

The proof is proposed in Appendix D. Theorem 2 justifies that the generalization gap of spiking GDA relies on the domain divergence $2C_f C_g W_1(\mathbb{P}_S(G), \mathbb{P}_T(G))$ and model discriminability $\omega$, as well as the model's ability to avoid overfitting to the training data, which is quantified by the Rademacher complexity term $2\mathbb{E}\left[\sup \frac{1}{N_S}\sum_{i=1}^{N_S} \epsilon_i h(\mathbf{X}_i, y_i, p_i)\right]$. In the application of spiking GDA, this term captures how the model's sensitivity to random fluctuations in the node feature aggregation (especially for higher-degree nodes) can lead to overfitting, thus affecting the model's ability to generalize to the target domain. This overfitting risk is particularly relevant when the model is too flexible in fitting the training data, exacerbating the generalization gap, especially under domain shifts.

### 4.3 PSEUDO-LABEL DISTILLATION FOR DISCRIMINATION LEARNING

To further tiny the generalization gap between the target and source domains, we incorporate the pseudo-label distillation module into the DeSGraDA framework. The goal of the module is to ensure consistent prediction between the shallow and deep layers. Specifically, let $\mathbf{s}'^t_{\tau, G_i}$ be the shallow spiking graph representation of $G_i$ on the latency step $\tau$ $(\tau < T)$ in the target domain, and $\hat{y}^t_{G_i} = H(\mathbf{s}_{G_i})$ be the prediction of graph $G_i$. Then, to enhance consistency between the shallow and deep feature spaces and facilitate the generation of more accurate predictions, we cluster the shallow graphs features in the target domain into $C$ clusters, and each cluster $\mathcal{E}_j$ includes graphs $\{G_j^t\}$. After that, we find the dominating labels $e_r$ in the cluster, i.e., $\max_r |\{\mathcal{E}_r : e_r = \hat{y}_{G_j}^t\}|$, and remove other samples with the same prediction but in different clusters. Formally, the pseudo-labels are signed as:

$$\mathcal{P} = \left\{ \left(G_j^t, \hat{y}_{G_j}^t\right) : e_j = \max_r \left|\left\{\mathcal{E}_r : e_r = \hat{y}_{G_j}^t\right\}\right| \right\}. \tag{7}$$

Finally, we utilize the distilled pseudo-labels to guide the update of source degree thresholds on the target domain with Eq. 3, and to direct classification in the target domain:

$$\mathcal{L}_T = \mathbb{E}_{G_j^t \in \mathcal{P}} l\left(H(\mathbf{s}_{G_j^t}^t), \hat{y}_{G_j^t}^t\right), \tag{8}$$

where $\mathbf{s}_{G_j^t}^t$ is the spiking graph representation of $G_j^t$ in the target domain. $l(\cdot)$ is the loss function, and we implement it with cross-entropy loss. We further analyze the generalization bound by applying the pseudo-label distillation module, and the proof is detailed in Appendix E. From the proof, we observe that the bound is lower than simply aligning the distributions by incorporating the highly reliable pseudo-labels, demonstrating the effectiveness of pseudo-labels for spiking graph domain adaptation.

### 4.4 LEARNING FRAMEWORK

Overall, the training objective of DeSGraDA integrates classification loss $\mathcal{L}_S$, temporal-based distribution alignment loss $\mathcal{L}_{AD}$, and pseudo-label distillation loss $\mathcal{L}_T$, which is formulated as:

$$\mathcal{L} = \mathcal{L}_S + \mathcal{L}_T - \lambda\mathcal{L}_{AD}, \tag{9}$$

where $\lambda$ is a hyper-parameter to balance the distribution alignment loss and classification loss. The learning procedure is illustrated in Algorithm F, and the complexity is shown in Appendix G.

Table 1: The graph classification results (in %) on SEED and BCI under edge density domain shift (source→target). S0, S1, S2, B0, B1, and B2 denote the sub-datasets of SEED and BCI partitioned with edge density, respectively. **Bold** results indicate the best performance.

| Methods | SEED | | | | | | BCI | | | | | |
|---|---|---|---|---|---|---|---|---|---|---|---|---|
| | S0→S1 | S1→S0 | S0→S2 | S2→S0 | S1→S2 | S2→S1 | B0→B1 | B1→B0 | B0→B2 | B2→B0 | B1→B2 | B2→B1 |
| WL subtree | 40.7 | 36.8 | 42.6 | 35.0 | 35.6 | 38.3 | 47.9 | 47.7 | 46.0 | 46.7 | 47.7 | 47.5 |
| GCN | 46.5±0.6 | 47.4±0.9 | 46.6±1.2 | 47.7±1.4 | 45.8±1.2 | 47.1±1.6 | 49.6±2.5 | 48.7±2.8 | 51.1±1.0 | 51.5±2.0 | 49.6±2.4 | 49.1±1.7 |
| GIN | 47.4±1.7 | 48.0±1.6 | 47.3±1.4 | 47.5±1.8 | 41.6±2.0 | 46.1±1.3 | 49.4±2.5 | 48.4±2.1 | 51.8±1.4 | 51.2±2.5 | 50.0±1.7 | 48.7±2.1 |
| GMT | 46.5±0.5 | 47.8±1.3 | 47.2±0.7 | 47.2±1.3 | 46.4±0.9 | 46.4±1.2 | 48.8±1.3 | 47.8±1.1 | 49.4±1.0 | 48.5±1.5 | 50.7±0.9 | 51.5±1.5 |
| CIN | 46.9±0.5 | 48.4±1.1 | 47.0±1.5 | 47.3±0.7 | 47.0±1.6 | 47.0±0.9 | 50.3±1.6 | 48.8±1.5 | 50.4±1.3 | 50.4±1.2 | 50.1±1.5 | 50.9±1.7 |
| SpikeGCN | 46.3±1.0 | 47.4±0.8 | 45.8±1.2 | 47.7±1.1 | 45.8±1.5 | 46.4±1.2 | 52.5±1.6 | 52.8±1.3 | 54.1±1.9 | 52.1±1.0 | 51.7±1.8 | 50.5±2.3 |
| DRSGNN | 47.1±1.0 | 48.5±0.9 | 46.5±1.2 | 48.1±1.3 | 46.9±0.8 | 47.6±1.4 | 52.7±1.3 | 52.8±1.9 | 53.8±1.5 | 52.7±1.6 | 53.0±2.1 | 51.3±1.8 |
| CDAN | 52.6±1.2 | 54.5±0.7 | 53.9±0.7 | 55.9±1.3 | 51.6±1.1 | 53.6±0.8 | 51.9±1.3 | 52.6±1.4 | 51.8±1.1 | 55.4±1.8 | 52.5±1.5 | 53.1±1.4 |
| ToAlign | 51.2±1.3 | 52.3±0.8 | 51.5±0.9 | 49.7±1.5 | 49.6±1.1 | 49.4±1.3 | 52.5±1.7 | 53.7±1.5 | 52.2±1.4 | 54.4±1.2 | 52.7±1.0 | 51.8±1.3 |
| MetaAlign | 51.2±1.4 | 53.7±0.9 | 52.2±1.1 | 53.8±0.8 | 51.2±1.4 | 52.0±1.2 | 51.1±1.5 | 51.8±1.2 | 50.4±1.7 | 52.5±1.5 | 51.7±1.5 | 51.3±1.1 |
| DEAL | 57.4±1.1 | 57.5±1.4 | 56.6±0.7 | 58.1±1.2 | 53.9±0.7 | 57.8±1.3 | 53.7±1.4 | 52.5±2.2 | 52.6±1.6 | 54.5±1.4 | 52.7±1.7 | 52.8±1.2 |
| CoCo | 55.5±1.5 | 56.7±0.7 | 56.3±1.3 | **58.8±0.8** | 54.2±1.2 | 57.5±1.3 | 54.0±1.3 | **55.2±2.5** | 52.7±2.1 | 52.7±1.9 | 51.7±2.8 | 51.0±2.4 |
| SGDA | 47.1±0.6 | 41.6±1.4 | 43.8±0.7 | 45.9±1.2 | 49.4±1.1 | 50.1±1.5 | 49.7±1.6 | 48.4±1.5 | 50.6±1.0 | 50.4±1.3 | 50.5±1.2 | 50.7±1.4 |
| StruRW | 47.1±0.9 | 45.9±0.7 | 46.5±1.3 | 48.2±1.2 | 46.9±1.2 | 47.3±1.4 | 48.7±1.1 | 47.3±1.7 | 49.5±1.1 | 49.7±1.5 | 50.0±1.8 | 50.2±1.6 |
| A2GNN | 47.6±1.2 | 47.6±0.9 | 46.2±0.8 | 48.3±1.1 | 46.2±1.0 | 47.9±0.6 | 52.0±1.7 | 53.0±1.4 | 52.0±1.0 | 53.7±1.1 | 52.2±1.3 | 51.8±1.7 |
| PA-BOTH | 48.2±1.4 | 48.2±0.8 | 47.3±1.2 | 48.3±1.0 | 48.5±1.2 | 45.2±0.6 | 49.2±1.6 | 50.0±1.2 | 51.1±1.3 | 51.3±1.5 | 50.5±1.6 | 48.8±1.4 |
| DeSGraDA | **58.0±1.5** | **58.2±1.4** | **57.0±1.8** | 58.3±1.6 | **55.9±2.1** | **58.1±1.6** | **54.1±1.5** | 53.6±1.6 | **54.9±1.1** | **56.2±1.8** | **55.0±1.3** | **54.6±1.2** |

## 5 EXPERIMENT

### 5.1 EXPERIMENTAL SETTINGS

**Dataset.** To evaluate the effectiveness of DeSGraDA, we conduct extensive experiments across two types of domain shifts: (1) structure-based domain shifts, where the discrepancy between domains arises primarily from differences in graph topology, such as variations in node and edge densities. This category includes datasets DD, PROTEINS (Dobson & Doig, 2003), SEED Zheng & Lu (2015); Duan et al. (2013), and BCI (Brunner et al., 2008); (2) feature-based domain shifts, where domains differ mainly in semantic information. This setting includes DD, PROTEINS, BZR, BZR_MD, COX2, and COX2_MD (Dobson & Doig, 2003; Sutherland et al., 2003). The specific statistics, distribution visualization, and detailed introduction of experimental datasets are presented in Appendix H.

**Baselines.** We compare DeSGraDA with competitive baselines on above datasets, including one graph kernel method: WL subtree (Shervashidze et al., 2011); four graph-based methods: GCN (Kipf & Welling, 2017), GIN (Xu et al., 2018), CIN (Bodnar et al., 2021) and GMT (Baek et al., 2021); two spiking-based graph methods: SpikeGCN (Zhu et al., 2022) and DRSGNN (Zhao et al., 2024); three domain adaptation methods: CDAN (Long et al., 2018), ToAlign (Wei et al., 2021b), and MetaAlign (Wei et al., 2021a); and six graph domain adaptation methods: DEAL (Yin et al., 2022), CoCo (Yin et al., 2023), SGDA (Qiao et al., 2023), StruRW (Liu et al., 2023), A2GNN (Liu et al., 2024a) and PA-BOTH (Liu et al., 2024b). More settings about baselines are introduced in Appendix I and J.

### 5.2 PERFORMANCE COMPARISON

We present the results of the proposed DeSGraDA with all baselines under two types of domain shifts on different datasets in Tables 1, 2, and 13-16. From these tables, we observe that: (1) GDA methods outperform traditional graph-based and spiking-based graph methods in most cases, highlighting the adverse impact of domain distribution shifts on conventional approaches and underscoring the impor-

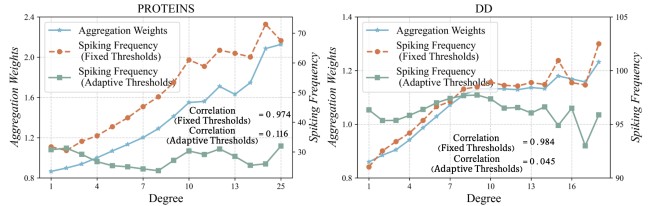

Figure 3: Correlation comparisons of spiking frequency and aggregation weights under adaptive thresholds on PROTEINS and DD datasets.

tance of advancing research in spiking graph domain adaptation. (2) The spiking-based graph methods (i.e., SpikeGCN and DRSGNN) outperform the models specific for node classification (i.e., SGDA, StruRW, A2GNN, and PA-BOTH) but fall short compared to models for graph classification (i.e., DEAL and CoCo). This performance gap is primarily due to the limited exploration of graph classification under domain shift. Although spiking-based methods exhibit advantages over adapted

Table 2: Graph classification results (in %) under node and edge density domain shifts on the PROTEINS dataset, and feature domain shifts on DD, PROTEINS, BZR, BZR_MD, COX2, and COX2_MD. For convenience, PROTEINS, DD, COX2, COX2_MD, BZR, and BZR_MD are abbreviated as P, D, C, CM, B, and BM, respectively. **Bold** results indicate the best performance.

| Methods | Node Shift | | | Edge Shift | | | Feature Shift | | | | | |
|---|---|---|---|---|---|---|---|---|---|---|---|---|
| | P0→P1 | P0→P2 | P0→P3 | P0→P1 | P0→P2 | P0→P3 | P→D | D→P | C→CM | CM→C | B→BM | BM→B |
| WL subtree | 69.1 | 61.2 | 41.6 | 68.7 | 50.7 | 58.1 | 43.0 | 42.2 | 53.1 | 58.2 | 51.3 | 44.0 |
| GCN | 73.7±0.3 | 57.6±0.2 | 24.4±0.4 | 73.4±0.2 | 57.6±0.2 | 24.0±0.1 | 48.9±2.0 | 60.9±2.3 | 51.2±1.8 | 66.9±1.8 | 48.7±2.0 | 78.8±1.7 |
| GIN | 71.8±2.7 | 58.5±4.3 | 74.2±1.7 | 62.5±4.7 | 53.0±4.6 | 73.7±0.8 | 57.3±2.2 | 61.9±1.9 | 53.8±2.5 | 55.6±2.0 | 49.9±2.4 | 79.2±2.8 |
| GMT | 73.7±0.2 | 57.6±0.3 | 75.6±1.4 | 73.4±0.3 | 57.6±0.1 | 24.0±0.1 | 59.5±2.5 | 50.7±2.2 | 49.3±1.8 | 58.2±2.0 | 50.2±2.3 | 74.4±1.8 |
| CIN | 74.1±0.6 | 60.1±2.1 | 75.6±0.2 | 74.5±0.2 | 57.8±0.2 | 75.6±0.6 | 59.1±2.6 | 58.0±2.7 | 51.2±2.0 | 55.6±1.5 | 49.2±1.4 | 74.2±1.9 |
| SpikeGCN | 71.8±0.9 | 64.9±1.4 | 71.1±1.9 | 71.8±0.8 | 63.8±1.0 | 68.6±1.1 | 59.6±2.2 | 63.3±1.8 | 52.6±2.5 | 68.6±1.8 | 53.3±1.7 | 76.1±2.0 |
| DRSGNN | 73.6±1.1 | 64.6±1.2 | 70.2±1.7 | 72.6±0.6 | 63.1±1.4 | 70.4±1.9 | 60.9±2.4 | 65.4±1.9 | 52.9±1.8 | 66.9±2.3 | 52.8±1.7 | 76.4±2.7 |
| CDAN | 75.9±1.0 | 60.8±0.6 | 75.8±0.3 | 72.2±1.8 | 59.8±2.1 | 69.3±4.1 | 59.4±2.0 | 63.1±2.7 | 51.2±2.3 | 68.2±1.8 | 50.7±1.6 | 75.2±1.9 |
| ToAlign | 73.7±0.4 | 57.6±0.6 | 24.4±0.1 | 73.4±0.1 | 57.6±0.1 | 24.0±0.3 | 62.1±2.1 | 66.5±2.3 | 53.2±2.6 | 55.8±2.3 | 56.2±3.0 | 78.8±2.4 |
| MetaAlign | 74.3±0.8 | 60.6±1.7 | 76.3±0.3 | 75.5±0.9 | 64.8±1.6 | 69.3±2.7 | 63.3±2.1 | 66.2±1.9 | 51.2±2.0 | 69.5±2.3 | 48.7±1.8 | 76.8±2.7 |
| DEAL | 75.4±1.2 | 68.1±1.9 | 73.8±1.4 | 76.5±0.4 | 67.5±1.3 | 76.0±0.2 | 70.6±1.9 | 66.8±2.5 | 50.9±2.4 | 67.8±1.9 | 51.1±2.3 | 79.4±2.2 |
| CoCo | 74.8±0.6 | 65.5±0.4 | 72.4±2.9 | 75.5±0.2 | 59.8±0.5 | 73.6±2.3 | 66.0±2.7 | 61.2±2.3 | 53.6±1.8 | 78.2±2.0 | **57.8±1.6** | 79.8±1.8 |
| SGDA | 64.2±0.5 | 66.9±1.2 | 65.4±1.6 | 63.8±0.6 | 66.7±1.0 | 60.1±0.8 | 48.3±2.0 | 55.8±2.6 | 49.8±1.8 | 66.9±2.3 | 50.3±2.1 | 78.8±2.6 |
| A2GNN | 65.7±0.6 | 66.3±0.9 | 65.2±1.4 | 65.4±1.3 | 66.2±2.2 | 65.4±0.7 | 57.8±2.1 | 60.3±1.5 | 51.5±1.8 | 67.7±2.1 | 51.6±2.3 | 77.5±1.9 |
| StruRW | 71.9±2.3 | 66.7±1.8 | 52.8±1.9 | 72.6±2.2 | 66.2±2.2 | 48.9±2.0 | 59.1±2.3 | 58.8±2.8 | 51.2±2.0 | 54.8±2.9 | 49.2±1.4 | 74.7±2.1 |
| PA-BOTH | 61.0±0.8 | 60.3±0.6 | 63.7±1.5 | 63.1±0.7 | 64.3±0.5 | 66.3±0.7 | 54.2±3.2 | 56.7±2.6 | 52.9±2.8 | 61.8±2.0 | 47.5±3.0 | 78.8±1.9 |
| DeSGraDA | **76.3±1.9** | **69.2±2.3** | **77.5±2.2** | **76.8±1.9** | **68.6±1.8** | **76.5±2.8** | **73.6±1.9** | **71.2±1.6** | **54.7±1.8** | **78.6±2.2** | 56.3±1.5 | **80.3±1.9** |

node classification models, they remain less effective than specialized graph domain adaptation methods explicitly designed for graph-level tasks. (3) DeSGraDA outperforms all baselines in most cases, demonstrating its advantage over other methods. The superior performance can be attributed to two key factors. First, the degree-conscious spiking representations dynamically adjust node-specific firing thresholds in SNNs, enabling the model to capture more expressive and discriminative graph features. Second, the temporal-based distribution alignment aligns source and target domain representations by matching spiking membrane dynamics over time, effectively mitigating distributional discrepancies. Moreover, the pseudo-label distillation helps refine degree thresholds in the target domain, further enhancing generalization. More results can be found in Appendix K.1.

We further conduct experiments to examine the correlation between spiking frequency and aggregation weights under adaptive thresholds across different node degrees. As shown in Figure 3, the correlation coefficient under adaptive thresholds is significantly lower than that under fixed thresholds, demonstrating that DeSGraDA effectively smooths spiking frequency across node degrees, mitigates over-activation in high-degree nodes, and promotes balanced information aggregation.

Table 3: The results of ablation studies on the PROTEINS dataset (source → target).

| Methods | P0→P1 | P1→P0 | P0→P2 | P2→P0 | P0→P3 | P3→P0 | P1→P2 | P2→P1 | P1→P3 | P3→P1 | P2→P3 | P3→P2 |
|---|---|---|---|---|---|---|---|---|---|---|---|---|
| DeSGraDA w/ CDAN | 73.3 | 82.4 | 67.8 | 76.5 | 74.4 | 78.7 | 66.5 | 71.3 | 73.7 | 70.2 | 74.1 | 68.8 |
| DeSGraDA w/o PL | 73.7 | 81.2 | 67.1 | 81.2 | 75.9 | 79.6 | 67.8 | 71.9 | 74.7 | 69.0 | 75.4 | 68.3 |
| DeSGraDA w/o CF | 65.0 | 67.4 | 53.5 | 64.5 | 66.6 | 68.8 | 56.7 | 63.4 | 66.1 | 60.9 | 53.9 | 56.6 |
| DeSGraDA w/o TL | 73.6 | 80.6 | 65.6 | 80.6 | 73.1 | 78.4 | 63.9 | 69.3 | 69.6 | 68.7 | 72.9 | 64.5 |
| DeSGraDA w/ CDAN & w/o PL | 72.8 | 81.0 | 66.3 | 76.3 | 72.7 | 77.6 | 65.8 | 69.6 | 71.0 | 68.8 | 73.6 | 67.8 |
| DeSGraDA w/ CDAN & w/o TL | 73.1 | 80.3 | 65.2 | 75.9 | 72.5 | 78.0 | 63.9 | 69.0 | 69.5 | 68.2 | 73.3 | 64.9 |
| DeSGraDA w/ CDAN, w/o PL & TL | 71.4 | 78.7 | 63.5 | 74.2 | 71.2 | 76.2 | 62.3 | 68.0 | 68.4 | 67.3 | 70.5 | 63.0 |
| DeSGraDA w/o PL & TL | 72.2 | 79.9 | 65.4 | 78.6 | 73.1 | 77.7 | 63.0 | 69.9 | 69.0 | 68.0 | 72.4 | 63.7 |
| DeSGraDA | **76.3** | **84.6** | **69.2** | **83.6** | **77.5** | **83.7** | **69.8** | **74.0** | **76.2** | **73.0** | **77.8** | **70.5** |

## 5.3 ABLATION STUDY

We conduct comprehensive ablation studies to assess the contribution of each component: (1) DeSGraDA w/ CDAN: replaces the temporal-based alignment module with static distribution alignment; (2) DeSGraDA w/o PL: removes the pseudo-label distillation module; (3) DeSGraDA w/o CF: discards the classification loss $\mathcal{L}_S$; (4) DeSGraDA w/o TL: applies fixed global thresholds to all nodes; (5) DeSGraDA w/ CDAN & w/o PL: employs static distribution alignment and removes pseudo-label distillation; (6) DeSGraDA w/ CDAN & w/o TL: adopts static alignment while using fixed global thresholds; (7) DeSGraDA w/ CDAN, w/o PL & TL: eliminates all three modules, retaining only the backbone with fixed thresholds; and (8) DeSGraDA w/o PL & TL: removes pseudo-label distillation and applies fixed thresholds.

Experimental results are shown in Table 3. From the table, we observe that: (1) The degree-conscious thresholding mechanism substantially improves representational capacity. When replaced with fixed global thresholds (DeSGraDA w/o TL), performance declines, showing that dynamically adjusting thresholds by node degree helps the model capture structural heterogeneity and encode informative

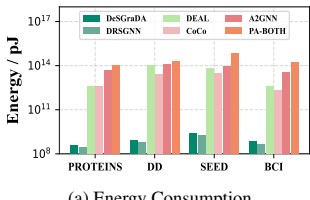
(a) Energy Consumption

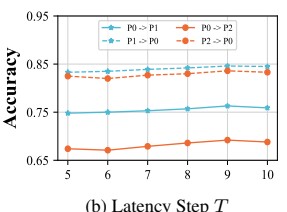
(b) Latency Step $T$

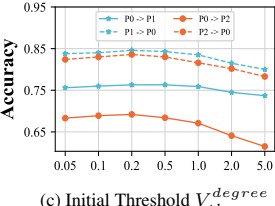
(c) Initial Threshold $V_{th}^{degree}$

Figure 4: (a) Energy efficiency analysis and (b), (c) hyperparameter sensitivity analysis of latency step $T$ and initial threshold $V_{th}^{degree}$ on the PROTEINS dataset.

spiking representations. (2) The temporal-based distribution alignment and pseudo-label distillation modules are crucial for domain adaptation. Removing the temporal alignment module (DeSGraDA w/ CDAN) yields consistent drops across tasks, underscoring its role in mitigating domain shifts by aligning membrane dynamics. Similarly, eliminating pseudo-label distillation (DeSGraDA w/o PL) degrades performance, highlighting the importance of leveraging confident target predictions to refine threshold adaptation and support generalization. (3) When multiple components are removed simultaneously, performance drops more substantially across all transfer tasks, suggesting that each module contributes both independently and cooperatively to the overall effectiveness of the framework. This finding highlights that the degree-conscious representation, temporal alignment, and pseudo-label distillation modules function complementarily to strengthen the robustness of domain adaptation. (4) DeSGraDA outperforms all ablated variants across domain shifts, confirming the complementary strengths of its core components. Notably, removing the classification loss (DeSGraDA w/o CF) causes the largest degradation, underscoring the necessity of source supervision for learning discriminative features. We also provide ablation studies replacing SGNs with standard GNNs in Tables 8, 9, with results in Appendix K.3.

## 5.4 ENERGY EFFICIENCY ANALYSIS

To assess the energy efficiency of DeSGraDA, we use the metric from (Zhu et al., 2022) and quantify the energy consumption for graph classification in the inference stage. Specifically, the graph domain adaptation methods are evaluated on GPUs (NVIDIA A100), and the spiking-based methods are evaluated on neuromorphic chips (ROLLS (Indiveri et al., 2015)) following (Zhu et al., 2022). The results are shown in Figure 4a, where we find that compared with traditional graph domain adaptation methods, the spike-based methods (DeSGraDA and DRSGNN) have significantly lower energy consumption, demonstrating the superior energy efficiency of SGNs. Moreover, although the energy consumption of DeSGraDA is slightly higher than DRSGNN due to additional computations required for domain adaptation, the performance improvement justifies deploying DeSGraDA in low-power devices. Additionally, we present a comparison of training time and memory usage between DeSGraDA and other GDA methods, and the results are detailed in Tables 6 and 7.

## 5.5 SENSITIVITY ANALYSIS

We conduct the sensitivity analysis of DeSGraDA to investigate the impact of key hyperparameters: latency step $T$ and degree threshold $V_{th}^{degree}$ in SGNs. Specifically, $T$ controls the number of SGNs propagation steps, and $V_{th}^{degree}$ determines the firing threshold of each neuron based on node degree.

Figure 4b and 4c illustrates how $T$ and $V_{th}^{degree}$ affects the performance of DeSGraDA on the PROTEINS dataset. More results on other datasets are shown in Appendix K.4. We vary $T$ in $\{5, 6, 7, 8, 9, 10\}$ and $V_{th}^{degree}$ in $\{0.05, 0.1, 0.2, 0.5, 1.0, 2.0, 5.0\}$. From the results, we observe that: (1) The performance of DeSGraDA in Figure 4b generally exhibits an increasing trend at the beginning and then stabilizes when $T > 9$. We attribute this to smaller values of $T$ potentially losing important information for representation, while larger values significantly increase model complexity. To balance effectiveness and efficiency, we set $T = 9$ as default. (2) Figure 4c indicates an initial increase followed by a decreasing trend in performance as $V_{th}^{degree}$ increases. This trend arises because a lower threshold may cause excessive spiking for high-degree nodes, leading to unstable representation, while a higher threshold may suppress spiking for low-degree nodes, reducing information flow. Accordingly, we set $V_{th}^{degree}$ to 0.2 as default.

Table 4: Wasserstein distance ($W_1$) between source and target domains before and after adaptation under node/edge density shifts (PROTEINS) and structure-based shifts (SEED, BCI).

| Method | Node Density (PROTEINS) | | | Edge Density (PROTEINS) | | | SEED | | | BCI | | |
|---|---|---|---|---|---|---|---|---|---|---|---|---|
| | P0→P1 | P0→P2 | P0→P3 | P0→P1 | P0→P2 | P0→P3 | S0→S1 | S0→S2 | S1→S2 | B0→B1 | B0→B2 | B1→B2 |
| Before | 0.0087 | 0.0187 | 0.0081 | 0.0089 | 0.0254 | 0.0054 | 0.0052 | 0.0047 | 0.0053 | 0.0044 | 0.0053 | 0.0053 |
| After | 0.0082 | 0.0172 | 0.0077 | 0.0086 | 0.0242 | 0.0053 | 0.0050 | 0.0045 | 0.0049 | 0.0043 | 0.0051 | 0.0050 |

## 5.6 EMPIRICAL VALIDATION OF THEORETICAL ANALYSIS

Theorem 2 provides a generalization bound for SGDA, which describes the target-domain risk $\epsilon_T$ as:

$$
\epsilon_T(h, \hat{h}) \leq \hat{\epsilon}_S(h, \hat{h}) + \underbrace{2C_f C_g W_1(P_S(G), P_T(G))}_{\text{domain divergence term}} + \underbrace{\omega}_{\substack{\text{model} \\ \text{discriminability}}} + \underbrace{\mathcal{O}\left(\sqrt{\frac{\ln(1/\delta)}{N_S}}\right)}_{\text{sample bound}} + \underbrace{\mathcal{R}(h)}_{\substack{\text{complexity} \\ \text{term}}}, \quad (10)
$$

where $W_1(P_S, P_T)$ denotes the distance between graph domain distributions, $\omega$ measures discriminator alignment, and $\mathcal{R}(h)$ represents model complexity related to the spiking activation dynamics.

However, directly computing this bound is infeasible because both $W_1(P_S, P_T)$ and $\mathcal{R}(h)$ depend on the underlying spiking feature distributions and unknown Lipschitz constants. This makes the bound unobservable in practice, as is common in theoretical domain adaptation and SNN analyses. Following prior works (Shen et al., 2018; Redko et al., 2017; Maass, 1997; Zhu et al., 2022), we validate our theory through the empirical consistency of trends predicted by Eq. (10) rather than through direct numerical computation from the follow parts:

**Domain Divergence Term.** To validate the theoretical domain divergence term, we computed the Wasserstein distance between source and target domains before and after applying our alignment. As shown in Table 4, the distance decreases after adaptation, indicating that the temporal alignment module effectively narrows the distributional gap. This trend aligns with Theorem 2, where a smaller $W_1$ implies a smaller generalization gap. These results provide empirical evidence that DeSGraDA achieves the theoretically predicted domain alignment.

**Rademacher Complexity and Model Capacity.** The Rademacher complexity term reflects overfitting risk from degree-induced activation bias. Figures 2 and 3 empirically verify that the Degree-Conscious mechanism reduces spiking over-activation. The correlation between node degree and spike frequency is strong before adaptation but notably weaker afterward, showing that the model avoids overfitting to structural patterns. This observation supports the theoretical claim that controlling model complexity (via reduced Rademacher complexity) enhances generalization stability.

**Target Risk $\epsilon_T$ and Domain Shift.** Tables 1 and 2 show that DeSGraDA consistently outperforms all baselines under both structure and feature domain shifts. These results empirically support Theorem 2: as the domain discrepancy decreases through alignment, the target risk $\epsilon_T$ also decreases. Conversely, when the shift intensifies, baselines show a larger rise in $\epsilon_T$, whereas DeSGraDA remains stable, confirming the bound's monotonic relationship between divergence and target error.

Overall, the empirical results provide comprehensive validation of the theoretical framework from multiple perspectives, including distributional alignment, complexity control, and generalization behavior. The consistent trends across datasets and experimental settings confirm that the theoretical analysis of DeSGraDA is mathematically rigorous and well supported by experimental evidence.

## 6 CONCLUSION

In this paper, we propose the problem of spiking graph domain adaptation and introduce a novel framework DeSGraDA for graph classification. DeSGraDA enhances the adaptability and performance of SGNs through three key aspects: degree-conscious spiking representation, temporal distribution alignment, and pseudo-label distillation. DeSGraDA captures expressive information via degree-dependent spiking thresholds, aligns feature distributions through temporal dynamics, and effectively exploits unlabeled target data via pseudo-label refinement. Extensive experiments on benchmark datasets demonstrate that DeSGraDA surpasses existing methods in accuracy while maintaining energy efficiency, showcasing its potential as a strong solution for DA in SGNs. In the future, we plan to extend DeSGraDA to source-free and domain-generalization scenarios.

ETHICS STATEMENT

This work complies with the ICLR Code of Ethics. We present DeSGraDA, a framework for spiking graph domain adaptation, evaluated on publicly available benchmark datasets. These datasets contain no personally identifiable or sensitive information, ensuring no risks to privacy or security. Our research advances energy-efficient graph learning with potential benefits for scientific and technological applications. All experimental protocols are transparently documented, with fair comparisons to prior work. The contributions are intended solely for research, supporting AI development.

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

APPENDICES

## A DA BOUND FOR GRAPHS

**DA Bound for Graphs.** Due to the DA theory is agnostic to data structures and encoders, You et al. (2023) directly rewrite it for graph-structured data ($G$) accompanied with graph feature extractors ($f$) as follows, and the covariate shift assumption is reframed as $\mathbb{P}_S(Y|G) = \mathbb{P}_T(Y|G)$.

**Theorem 3** *(You et al., 2023) Assume that the learned discriminator is $C_g$-Lipschitz continuous as in (Redko et al., 2017), and the graph feature extractor $f$ is $C_f$-Lipschitz that $||f||_{Lip} = \max_{G_1,G_2} \frac{||f(G_1)-f(G_2)||_2}{\eta(G_1,G_2)} = C_f$ for some graph distance measure $\eta$. Let $\mathcal{H} := \{h : \mathcal{G} \to \mathcal{Y}\}$ be the set of bounded real-valued functions with pseudo-dimension $Pdim(\mathcal{H}) = d$ that $h = g \circ f \in \mathcal{H}$, with probability at least $1 - \delta$ the following inequality holds:*

$$\epsilon_T(h, \hat{h}) \leq \hat{\epsilon}_S(h, \hat{h}) + \sqrt{\frac{4d}{N_S} \log\left(\frac{eN_S}{d}\right) + \frac{1}{N_S} \log\left(\frac{1}{\delta}\right)} + 2C_f C_g W_1(\mathbb{P}_S(G), \mathbb{P}_T(G)) + \omega,$$

*where $\epsilon_T(h, \hat{h}) = \mathbb{E}_{\mathbb{P}_T(G}\{|h(G) - \hat{h}(G)|\}$ is the (empirical) target risk, $\hat{\epsilon}_S(h, \hat{h}) = \frac{1}{N_s} \sum_{n=1}^{N_S} |h(G_n) - \hat{h}(G_n)|$ is the (empirical) source risk, $\hat{h} : \mathcal{G} \to \mathcal{Y}$ is the labeling function for graphs and $\omega = \min_{||g||_{Lip} \leq C_g, ||f||_{Lip} \leq C_f} \{\epsilon_S(h, \hat{h}) + \epsilon_T(h, \hat{h})\}$, and $W_1(\mathbb{P}, \mathbb{Q}) = \sup_{||g||_{Lip} \leq 1} \{\mathbb{E}_{\mathbb{P}_S(Z)} g(Z) - \mathbb{E}_{\mathbb{P}_T(Z)} g(Z)\}$ is the first Wasserstein distance (Villani et al., 2008).*

## B SPIKING NEURAL NETWORKS

Spiking Neural Networks (SNNs) are brain-inspired models that communicate through discrete spike events rather than continuous-valued activations Maass (1997); Gerstner & Kistler (2002); Bohte et al. (2000). This design provides significant advantages in temporal information processing and energy efficiency. Different from traditional neural networks, SNNs emulate biological mechanisms such as membrane potential integration, threshold-triggered firing, and post-spike resetting Cao et al. (2015); Zhu et al. (2022). To capture these biological dynamics, SNNs employ neuron models that mathematically describe the temporal evolution of membrane potentials and the conditions for spike generation Lagani et al. (2023). A widely used neuron model in SNNs is the Leaky Integrate-and-Fire (LIF) model Tal & Schwartz (1997); Lansky & Ditlevsen (2008), which operates through three fundamental stages:

(1) Integrate: At each lantency step $t$, the membrane potential $V[t]$ is updated by integrating the input current $I[t]$ and applying a decay to the previous potential $V[t-1]$:

$$V[t] = \lambda V[t-1](1 - S[t-1]) + I[t] \tag{11}$$

where $\lambda \in (0, 1)$ is the decay factor that controls the leakage rate, and $S[t-1]$ is the binary spike indicator from the previous time step.

(2) Fire: A spike is emitted when the membrane potential exceeds the threshold $V_{th}$:

$$S[t] = H(V[t] - V_{th}), \tag{12}$$

where $H(\cdot)$ denotes the Heaviside step function, a non-differentiable function defined as:

$$H(x) = \begin{cases} 1, & x \geq 0, \\ 0, & \text{otherwise.} \end{cases} \tag{13}$$

(3) Reset: After a spike is emitted, the membrane potential is reset according to the following rule:

$$V[t] = (1 - S[t])V[t] + S[t]V_{reset}, \tag{14}$$

where $V_{reset}$ denotes the resting potential, typically set to zero.

Due to the non-differentiability of $H(\cdot)$, surrogate gradient methods are commonly employed to approximate its derivative during backpropagation, enabling gradient-based optimization in deep SNN architectures Bohte et al. (2000); Sun et al. (2024).

## C  PROOF OF PROPOSITION 1

Assuming that the node feature $h_i$ follows a normal distribution $\mathcal{N}(\mu, \sigma^2)$, then for each node in the graph, we follow the message-passing mechanism and have the information aggregation as:

$$h_i = h_i + \sum_{j \in N(i)} w_{ij} h_j, \tag{15}$$

where $N(i)$ is the neighbor set of node $i$. Therefore, we have the expectation:

$$\mathbb{E}(h_i) = \mathbb{E}(h_i) + \sum_{j \in N(i)} w_{ij} \mathbb{E}(h_j). \tag{16}$$

Since $\mathbb{E}(h_j) \sim \mathcal{N}(\mu, \sigma^2)$, we have:

$$\mathbb{E}(h_i) \sim \mathcal{N}\left( (1 + \sum_{j \in N(i)} w_{ij})\mu, (1 + \sum_{j \in N(i)} w_{ij})^2 \sigma^2 \right). \tag{17}$$

From the results, we observe that node $i$ follows a normal distribution with a mean of $(1 + \sum_{j \in N(i)} w_{ij})^2 \mu$, determined by the aggregated weights of its neighboring nodes. To provide a more intuitive understanding, we visualize the aggregated neighbor weights of GCN (Kipf & Welling, 2017) and GIN (Xu et al., 2018) in Figure 2. The results show that as the degree increases, the aggregated weights also increase progressively. Consequently, high-degree nodes tend to follow a normal distribution with a higher mean and variance. In other words, nodes with higher degrees accumulate greater signals, making them more likely to trigger spiking. Based on this, we propose assigning higher thresholds to high-degree nodes and lower thresholds to low-degree nodes.

## D  PROOF OF THEOREM 2

**Theorem 2** *Assuming that the learned discriminator is $C_g$-Lipschitz continuous as described in Theorem 3, the graph feature extractor $f$ (also referred to as GNN) is $C_f$-Lipschitz that $||f||_{Lip} = \max_{G_1,G_2} \frac{||f(G_1)-f(G_2)||_2}{\eta(G_1,G_2)} = C_f$ for some graph distance measure $\eta$ and the loss function bounded by $C > 0$. Let $\mathcal{H} := \{h : \mathcal{G} \to \mathcal{Y}\}$ be the set of bounded real-valued functions with the pseudo-dimension $Pdim(\mathcal{H}) = d$ that $h = g \circ f \in \mathcal{H}$, and provided the spike training data set $S_n = \{(\mathbf{X}_i, y_i) \in \mathcal{X} \times \mathcal{Y}\}_{i \in [n]}$ drawn from $\mathcal{D}^s$, with probability at least $1 - \delta$ the following inequality:*

$$\epsilon_T(h, \hat{h}_T(\mathbf{X})) \leq \hat{\epsilon}_S(h, \hat{h}_S(\mathbf{S})) + 2\mathbb{E}\left[ \sup \frac{1}{N_S} \sum_{i=1}^{N_S} \epsilon_i h(\mathbf{X}_i, y_i, p_i) \right] + C\sqrt{\frac{ln(2/\delta)}{N_S}} + \omega + 2C_f C_g W_1\left( \mathbb{P}_S(G), \mathbb{P}_T(G) \right), \tag{18}$$

*where the (empirical) source and target risks are $\hat{\epsilon}_S(h, \hat{h}(\mathbf{S})) = \frac{1}{N_S} \sum_{n=1}^{N_S} |h(\mathbf{S}_n) - \hat{h}(\mathbf{S}_n)|$ and $\epsilon_T(h, \hat{h}(\mathbf{X})) = \mathbb{E}_{\mathbb{P}_T(G)}\{|h(G) - \hat{h}(G)|\}$, respectively, where $\hat{h} : \mathcal{G} \to \mathcal{Y}$ is the labeling function for graphs and $\omega = \min\left( |\epsilon_S(h, \hat{h}_S(\mathbf{X})) - \epsilon_S(h, \hat{h}_T(\mathbf{X}))|, |\epsilon_T(h, \hat{h}_S(\mathbf{X})) - \epsilon_T(h, \hat{h}_T(\mathbf{X}))| \right)$, $\epsilon_i$ is the Rademacher variable and $p_i$ is the $i^{th}$ row of $\mathbf{P}$, which is the probability matrix with:*

$$\mathbf{P}_{kt} = \begin{cases} \exp\left( \frac{u_k(t) - V_{th}}{\sigma(u_k(t) - u_{reset})} \right), & if \quad u_\theta \leq u(t) \leq V_{th}, \\ 0, & if \quad u_{reset} \leq u_k(t) \leq u_\theta. \end{cases} \tag{19}$$

*Proof.*

Before showing the designated lemma, we first introduce the following inequality to be used that:

$$\begin{aligned} |\epsilon_S(h, \hat{h}_S) - \epsilon_T(h, \hat{h}_T)| &= |\epsilon_S(h, \hat{h}_S) - \epsilon_S(h, \hat{h}_T) + \epsilon_S(h, \hat{h}_T) - \epsilon_T(h, \hat{h}_T)| \\ &\leq |\epsilon_S(h, \hat{h}_S) - \epsilon_S(h, \hat{h}_T)| + |\epsilon_S(h, \hat{h}_T) - \epsilon_T(h, \hat{h}_T)| \\ &\overset{(a)}{\leq} |\epsilon_S(h, \hat{h}_S) - \epsilon_S(h, \hat{h}_T)| + 2C_f C_g W_1\left( \mathbb{P}_S(G), \mathbb{P}_T(G) \right), \end{aligned} \tag{20}$$

where $(a)$ results from (Shen et al., 2018) Theorem 3 with the assumption $\max(||h||_{Lip}, \max_{G_1,G_2} \frac{|\hat{h}_D(G_1) - \hat{h}_D(G_2)|}{\eta(G_1,G_2)}) \leq C_f C_g, D \in \{S, T\}$. Similarly, we obtain:

$$|\epsilon_S(h, \hat{h}_S) - \epsilon_T(h, \hat{h}_T)| \leq |\epsilon_T(h, \hat{h}_S) - \epsilon_T(h, \hat{h}_T)| + 2C_f C_g W_1(\mathbb{P}_S(G), \mathbb{P}_T(G)). \tag{21}$$

We therefore combine them into:

$$\begin{aligned}
|\epsilon_S(h, \hat{h}_S) - \epsilon_T(h, \hat{h}_T)| \leq &2C_f C_g W_1(\mathbb{P}_S(G), \mathbb{P}_T(G)) \\
&+ \min\left(|\epsilon_S(h, \hat{h}_S) - \epsilon_S(h, \hat{h}_T)|, |\epsilon_T(h, \hat{h}_S) - \epsilon_T(h, \hat{h}_T)|\right),
\end{aligned} \tag{22}$$

i.e. the following holds to bound the target risk $\epsilon_T(h, \hat{h}_T)$:

$$\begin{aligned}
\epsilon_T(h, \hat{h}_T) \leq &\epsilon_S(h, \hat{h}_S) + 2C_f C_g W_1(\mathbb{P}_S(G), \mathbb{P}_T(G)) \\
&+ \min\left(|\epsilon_S(h, \hat{h}_S) - \epsilon_S(h, \hat{h}_T)|, |\epsilon_T(h, \hat{h}_S) - \epsilon_T(h, \hat{h}_T)|\right).
\end{aligned} \tag{23}$$

We next link the bound with the empirical risk and labeled sample size by showing, with probability at least $1 - \delta$ that:

$$\begin{aligned}
\epsilon_T(h, \hat{h}_T) \leq &\epsilon_S(h, \hat{h}_S) + 2C_f C_g W_1(\mathbb{P}_S(G), \mathbb{P}_T(G)) \\
&+ \min\left(|\epsilon_S(h, \hat{h}_S) - \epsilon_S(h, \hat{h}_T)|, |\epsilon_T(h, \hat{h}_S) - \epsilon_T(h, \hat{h}_T)|\right).
\end{aligned} \tag{24}$$

The $\hat{h}$ above is the abbreviation of $\hat{h}(x)$, which means the input is the continuous feature. Provided the spike training data set $S_n = \{(\mathbf{X}_i, y_i) \in \mathcal{X} \times \mathcal{Y}\}_{i \in [n]}$ drawn from $\mathcal{D}$, and motivated by (Yin et al., 2024), we have:

$$\lim_{\tau \to \infty} P\left(\hat{h}(S_n)_{\tau,i} > \hat{h}(\mathbf{X}_{\tau,i}) + \epsilon\right) \leq e^{-\epsilon^2/2(\sigma + \hat{w}_i \epsilon/3)}, \tag{25}$$

where $\hat{w}_i = max\{w_{i1}, \cdots, w_{id}\}$ and $h(\mathbf{x}_{ij}) = \sum_{j=1}^d w_{ij} \mathbf{x}_{ij}$. From Equation 2, we observe that as $\tau \to \infty$, the difference between spike and real-valued features will be with the probability of $p = e^{-\epsilon^2/2(\sigma + \hat{w}_i \epsilon/3)}$ to exceed the upper and lower bounds.

Furthermore, motivated by the techniques given by (Bartlett & Mendelson, 2002), we have:

$$\epsilon_S(h, \hat{h}_S(S_n)) \leq \hat{\epsilon}_S(h, \hat{h}_S(S_n)) + \underbrace{\sup[\epsilon_S(h, \hat{h}_S(S_n)) - \hat{\epsilon}_S(h, \hat{h}_S(S_n))]}_{R(S_n, \mathbf{P})}, \tag{26}$$

where $\mathbf{P}$ is the probability matrix with:

$$\mathbf{P}_{kt} = \begin{cases} \exp\left(\frac{u_k(t) - V_{th}}{\sigma(u_k(t) - u_{reset})}\right), & if \quad u_\theta \leq u(t) \leq V_{th}, \\ 0, & if \quad u_{reset} \leq u_k(t) \leq u_\theta, \end{cases} \tag{27}$$

where $k$ indicates the $k - th$ spiking neuron and the membrane threshold $u_{theta}$ is relative to the excitation probability threshold $p_\theta \in (0, 1]$. Let $p_k$ is the $k - th$ row vector of $\mathbf{P}$. Thus, we have the probability at least $1 - e^{-\epsilon^2/2(\sigma + \hat{w}_i \epsilon/3)}$ to hold:

$$\epsilon_S(h, \hat{h}_S(\mathbf{X}_n)) \leq \hat{\epsilon}_S(h, \hat{h}_S(S_n)) + \underbrace{\sup[\epsilon_S(h, \hat{h}_S(S_n)) - \hat{\epsilon}_S(h, \hat{h}_S(S_n))]}_{R(S_n, \mathbf{P})}, \tag{28}$$

Let $S'_n$ denote the sample set that the $i^{th}$ sample $(\mathbf{X}_i, y_i)$ is replaced by $(\mathbf{X}'_i, y'_i)$, and correspondingly $\mathbf{P}'$ is the possibility matrix that the $i^{th}$ row vector $p_i$ is replaced by $p'_i$, for $i \in [n]$. For the loss function bounded by $C > 0$, we have:

$$\begin{cases} |R(S_n, \mathbf{P}) - R(S'_n, \mathbf{P})| \leq C/n, \\ |R(S_n, \mathbf{P}) - R(S_n, \mathbf{P}')| \leq C/n. \end{cases} \tag{29}$$

From McDiarmid's inequality (McDiarmid et al., 1989), with probability at least $1 - \delta$, we have:

$$R(S_n, \mathbf{P}) \leq \mathbb{E}_{S_n \in \mathcal{D}, \mathbf{P}}[R(S_n, \mathbf{P})] + C\sqrt{\frac{ln(2/\delta)}{N_S}}. \tag{30}$$

It is observed that:

$$R(S_n, \mathbf{P}) = \sup \mathbb{E}_{\tilde{S}_n \in \mathcal{D}, \tilde{\mathbf{P}}}[\hat{\epsilon}(\hat{h}(S_n); \tilde{S}_n, \tilde{\mathbf{P}}) - \tilde{\mathbf{P}}[\hat{\epsilon}(\hat{h}(S_n); S_n, \mathbf{P})], \quad (31)$$

where $\tilde{S}_n$ is another collection drawn from $\mathcal{D}$ as well as $\tilde{\mathbf{P}}$. Thus, we have

$$\mathbb{E}_{S_n \in \mathcal{D}, \mathbf{P}}[R(S_n, \mathbf{P})] \leq \mathbb{E}\left[\sup\left[\hat{\epsilon}(\hat{h}(S_n); \tilde{S}_n, \tilde{\mathbf{P}}) - \tilde{\mathbf{P}}[\hat{\epsilon}(\hat{h}(S_n); S_n, \mathbf{P})]\right]\right]$$

$$= \mathbb{E}\left[\sup \frac{1}{n} \sum_{i=1}^n [\hat{h}(\tilde{\mathbf{X}}_i, \tilde{y}_i, \tilde{p}_i) - \hat{h}(\mathbf{X}_i, y_i, p_i)]\right] \quad (32)$$

$$\leq 2\mathbb{E}\left[\sup \frac{1}{n} \sum_{i=1}^n \epsilon_i \hat{h}(\mathbf{X}_i, y_i, p_i)\right],$$

where $\epsilon_i$ is the Rademacher variable. Combining Eq. 29 30 32, we have:

$$\epsilon_S(h, \hat{h}_S(\mathbf{X}_n)) \leq \hat{\epsilon}_S(h, \hat{h}_S(S_n)) + 2\mathbb{E}\left[\sup \frac{1}{N_S} \sum_{i=1}^{N_S} \epsilon_i h(\mathbf{X}_i, y_i, p_i)\right] + C\sqrt{\frac{ln(2/\delta)}{N_S}}. \quad (33)$$

Finally, we have:

$$\epsilon_T(h, \hat{h}_T(\mathbf{X})) \leq \epsilon_S(h, \hat{h}_S(\mathbf{X})) + 2C_f C_g W_1\left(\mathbb{P}_S(G), \mathbb{P}_T(G)\right)$$

$$+ \min\left(|\epsilon_S(h, \hat{h}_S(\mathbf{X})) - \epsilon_S(h, \hat{h}_T(\mathbf{X}))|, |\epsilon_T(h, \hat{h}_S(\mathbf{X})) - \epsilon_T(h, \hat{h}_T(\mathbf{X}))|\right)$$

$$\leq \hat{\epsilon}_S(h, \hat{h}_S(S_n)) + 2\mathbb{E}\left[\sup \frac{1}{N_S} \sum_{i=1}^{N_S} \epsilon_i h(\mathbf{X}_i, y_i, p_i)\right] + C\sqrt{\frac{ln(2/\delta)}{N_S}}$$

$$+ \min\left(|\epsilon_S(h, \hat{h}_S(\mathbf{X})) - \epsilon_S(h, \hat{h}_T(\mathbf{X}))|, |\epsilon_T(h, \hat{h}_S(\mathbf{X})) - \epsilon_T(h, \hat{h}_T(\mathbf{X}))|\right)$$

$$+ 2C_f C_g W_1\left(\mathbb{P}_S(G), \mathbb{P}_T(G)\right). \quad (34)$$

# E GENERALIZATION BOUND WITH PSEUDO-LABEL DISTILLATION MODULE

**Theorem 4** *Under the assumption of Theorem 3, we further assume that there exists a small amount of i.i.d. samples with pseudo labels $\{(G_n, Y_n)\}_{n=1}^{N_T'}$ from the target distribution $\mathbb{P}_T(G, Y)$ ($N_T' \ll N_S$) and bring in the conditional shift assumption that domains have different labeling function $\hat{h}_S \neq \hat{h}_T$ and $\max_{G_1, G_2} \frac{|\hat{h}_D(G_1) - \hat{h}_D(G_2)|}{\eta(G_1, G_2)} = C_h \leq C_f C_g (D \in \{S, T\})$ for some constant $C_h$ and distance measure $\eta$, and the loss function bounded by $C > 0$. Let $\mathcal{H} := \{h : \mathcal{G} \to \mathcal{Y}\}$ be the set of bounded real-valued functions with the pseudo-dimension $Pdim(\mathcal{H}) = d$, and provided the spike training data set $S_n = \{(\mathbf{X}_i, y_i) \in \mathcal{X} \times \mathcal{Y}\}_{i \in [n]}$ drawn from $\mathcal{D}^s$, with probability at least $1 - \delta$ the following inequality holds:*

$$\epsilon_T(h, \hat{h}_T(\mathbf{X})) \leq \frac{N_T'}{N_S + N_T'} \hat{\epsilon}_T(h, \hat{h}_T(S)) + \frac{N_S}{N_S + N_T'}\left(\hat{\epsilon}_S(h, \hat{h}_S(S)) + 2C_f C_g W_1\left(\mathbb{P}_S(G), \mathbb{P}_T(G)\right)\right.$$

$$+ 2\mathbb{E}\left[\sup \frac{1}{N_S} \sum_{i=1}^{N_S} \epsilon_i h(\mathbf{X}_i, y_i, p_i)\right] + C\sqrt{\frac{ln(2/\delta)}{N_S}} + \omega\right)$$

$$\leq Eq. 6, \quad (35)$$

*where the (empirical) source and target risks are $\hat{\epsilon}_S(h, \hat{h}) = \frac{1}{N_S} \sum_{n=1}^{N_S} |h(G_n) - \hat{h}(G_n)|$ and $\epsilon_T(h, \hat{h}) = \mathbb{E}_{\mathbb{P}_T(G}\{|h(G) - \hat{h}(G)|\}$, respectively, where $\hat{h} : \mathcal{G} \to \mathcal{Y}$ is the labeling function for graphs and $\omega = \min\left(|\epsilon_S(h, \hat{h}_S(\mathbf{X}))) - \epsilon_S(h, \hat{h}_T(\mathbf{X}))|, |\epsilon_T(h, \hat{h}_S(\mathbf{X}))) - \epsilon_T(h, \hat{h}_T(\mathbf{X}))|\right)$, $\epsilon_i$ is the Rademacher variable and $p_i$ is the $i^{th}$ row of $\mathbf{P}$, which is defined in Theorem 2.*

*Proof.*

As proved in Theorem 2, we have:

$$\epsilon_T(h, \hat{h}_T(\mathbf{X})) \leq \hat{\epsilon}_S(h, \hat{h}_S(S_n)) + 2\mathbb{E}\left[\sup \frac{1}{N_S}\sum_{i=1}^{N_S}\epsilon_i h(\mathbf{X}_i, y_i, p_i)\right] + C\sqrt{\frac{ln(2/\delta)}{N_S}}$$

$$+ \min\left(|\epsilon_S(h, \hat{h}_S(\mathbf{X})) - \epsilon_S(h, \hat{h}_T(\mathbf{X}))|, |\epsilon_T(h, \hat{h}_S(\mathbf{X})) - \epsilon_T(h, \hat{h}_T(\mathbf{X}))|\right)$$

$$+ 2C_f C_g W_1\left(\mathbb{P}_S(G), \mathbb{P}_T(G)\right). \tag{36}$$

Similar with Eq. 33, there exists:

$$\epsilon_T(h, \hat{h}_T(\mathbf{X}_n)) \leq \hat{\epsilon}_T(h, \hat{h}_T(S_n)) + 2\mathbb{E}\left[\sup \frac{1}{N_T'}\sum_{i=1}^{N_T'}\epsilon_i h(\mathbf{X}_i, y_i, p_i)\right] + C\sqrt{\frac{ln(2/\delta)}{N_T'}}. \tag{37}$$

Combining Eq. 36 and 37, we have:

$$\epsilon_T(h, \hat{h}_T(\mathbf{X})) \overset{(a)}{\leq} \frac{N_T'}{N_S + N_T'}\left(\hat{\epsilon}_T(h, \hat{h}_T(S)) + 2\mathbb{E}\left[\sup \frac{1}{N_T'}\sum_{i=1}^{N_T'}\epsilon_i h(\mathbf{X}_i, y_i, p_i)\right] + C\sqrt{\frac{ln(2/\delta)}{N_T'}}\right)$$

$$+ \frac{N_S}{N_S + N_T'}\left(\hat{\epsilon}_S(h, \hat{h}_S(S)) + 2\mathbb{E}\left[\sup \frac{1}{N_S}\sum_{i=1}^{N_S}\epsilon_i h(\mathbf{X}_i, y_i, p_i)\right] + C\sqrt{\frac{ln(2/\delta)}{N_S}}\right)$$

$$+ \frac{N_S}{N_S + N_T'}\left(2C_f C_g W_1\left(\mathbb{P}_S(G), \mathbb{P}_T(G)\right)\right.$$

$$\left. + \min\left(|\epsilon_S(h, \hat{h}_S(\mathbf{X})) - \epsilon_S(h, \hat{h}_T(\mathbf{X}))|, |\epsilon_T(h, \hat{h}_S(\mathbf{X})) - \epsilon_T(h, \hat{h}_T(\mathbf{X}))|\right)\right)$$

$$\leq \frac{N_T'}{N_S + N_T'}\hat{\epsilon}_T(h, \hat{h}_T(S)) + \frac{N_S}{N_S + N_T'}\hat{\epsilon}_S(h, \hat{h}_S(S))$$

$$+ \frac{N_S}{N_S + N_T'}\left(2C_f C_g W_1\left(\mathbb{P}_S(G), \mathbb{P}_T(G)\right)\right.$$

$$\left. + \min\left(|\epsilon_S(h, \hat{h}_S(\mathbf{X}) - \epsilon_S(h, \hat{h}_T((\mathbf{X}))|, |\epsilon_T(h, \hat{h}_S((\mathbf{X})) - \epsilon_T(h, \hat{h}_T((\mathbf{X}))|\right)\right)$$

$$+ \frac{N_T'}{N_S + N_T'}\left(2\mathbb{E}\left[\sup \frac{1}{N_T'}\sum_{i=1}^{N_T'}\epsilon_i h(\mathbf{X}_i, y_i, p_i)\right] + C\sqrt{\frac{ln(2/\delta)}{N_T'}}\right)$$

$$+ \frac{N_S}{N_S + N_T'}\left(2\mathbb{E}\left[\sup \frac{1}{N_S}\sum_{i=1}^{N_S}\epsilon_i h(\mathbf{X}_i, y_i, p_i)\right] + C\sqrt{\frac{ln(2/\delta)}{N_S}}\right)$$

$$\overset{(b)}{\doteq} \frac{N_T'}{N_S + N_T'}\hat{\epsilon}_T(h, \hat{h}_T(S)) + \frac{N_S}{N_S + N_T'}\hat{\epsilon}_S(h, \hat{h}_S(S))$$

$$+ \frac{N_S}{N_S + N_T'}\left(2\mathbb{E}\left[\sup \frac{1}{N_S}\sum_{i=1}^{N_S}\epsilon_i h(\mathbf{X}_i, y_i, p_i)\right] + C\sqrt{\frac{ln(2/\delta)}{N_S}}\right)$$

$$+ \frac{N_S}{N_S + N_T'} \left( 2C_f C_g W_1 \left( \mathbb{P}_S(G), \mathbb{P}_T(G) \right) \right.$$

$$+ \min \left( |\epsilon_S(h, \hat{h}_S(\mathbf{X})) - \epsilon_S(h, \hat{h}_T(\mathbf{X}))|, |\epsilon_T(h, \hat{h}_S(\mathbf{X})) - \epsilon_T(h, \hat{h}_T(\mathbf{X}))| \right) \right)$$

$$= \frac{N_T'}{N_S + N_T'} \hat{\epsilon}_T(h, \hat{h}_T(S)) + \frac{N_S}{N_S + N_T'} \left( \hat{\epsilon}_S(h, \hat{h}_S(S)) + 2C_f C_g W_1 \left( \mathbb{P}_S(G), \mathbb{P}_T(G) \right) \right.$$

$$+ 2\mathbb{E} \left[ \sup \frac{1}{N_S} \sum_{i=1}^{N_S} \epsilon_i h(\mathbf{X}_i, y_i, p_i) \right] + C \sqrt{\frac{ln(2/\delta)}{N_S}}$$

$$+ \min \left( |\epsilon_S(h, \hat{h}_S(\mathbf{X}))) - \epsilon_S(h, \hat{h}_T(\mathbf{X})))|, |\epsilon_T(h, \hat{h}_S(\mathbf{X}))) - \epsilon_T(h, \hat{h}_T(\mathbf{X})))| \right) \right)$$

where (a) is the outcome of applying the union bound with coefficient $\frac{N_T'}{N_S+N_T'}$, $\frac{N_S}{N_S+N_T'}$ respectively; (b) additionally adopt the assumption $N_T' \ll N_S$, following the sleight-of-hand in (Li et al., 2021) Theorem 3.2.

Due to the sampels are selected with high confidence, thus, we have the following assumption:

$$\hat{\epsilon}_T \leq \epsilon_T \leq \hat{\epsilon}_S(h, \hat{h}(\mathbf{X}))) + 2\mathbb{E} \left[ \sup \frac{1}{N_S} \sum_{i=1}^{N_S} \epsilon_i h(\mathbf{X}_i, y_i, p_i) \right]$$

$$+ C \sqrt{\frac{ln(2/\delta)}{N_S}} + 2C_f C_g W_1(\mathbb{P}_S(G), \mathbb{P}_T(G)) + \omega, \tag{38}$$

where $\omega = \min \left( |\epsilon_S(h, \hat{h}_S(\mathbf{X}))) - \epsilon_S(h, \hat{h}_T(\mathbf{X})))|, |\epsilon_T(h, \hat{h}_S(\mathbf{X}))) - \epsilon_T(h, \hat{h}_T(\mathbf{X})))| \right)$, $\hat{\epsilon}_T$ is the empirical risk on the high confidence samples, $\epsilon_T$ is the empirical risk on the target domain. Besides, we have:

$$\min(|\epsilon_S(h, \hat{h}_S(\mathbf{X}))) - \epsilon_S(h, \hat{h}_T(\mathbf{X})))|, |\epsilon_T(h, \hat{h}_S(\mathbf{X}))) - \epsilon_T(h, \hat{h}_T)|(\mathbf{X}))) \leq$$

$$\min \left( \epsilon_S(h, \hat{h}_S(\mathbf{X}))) + \epsilon_T(h, \hat{h}_S(\mathbf{X}))) \right) \tag{39}$$

Then,

$$\epsilon_T(h, \hat{h}_T(\mathbf{X})) \leq \frac{N_T'}{N_S + N_T'} \hat{\epsilon}_T(h, \hat{h}_T(S)) + \frac{N_S}{N_S + N_T'} \left( \hat{\epsilon}_S(h, \hat{h}_S(S)) + 2C_f C_g W_1 \left( \mathbb{P}_S(G), \mathbb{P}_T(G) \right) \right.$$

$$+ 2\mathbb{E} \left[ \sup \frac{1}{N_S} \sum_{i=1}^{N_S} \epsilon_i h(\mathbf{X}_i, y_i, p_i) \right] + C \sqrt{\frac{ln(2/\delta)}{N_S}}$$

$$+ \min \left( |\epsilon_S(h, \hat{h}_S(\mathbf{X}))) - \epsilon_S(h, \hat{h}_T(\mathbf{X})))|, |\epsilon_T(h, \hat{h}_S(\mathbf{X}))) - \epsilon_T(h, \hat{h}_T(\mathbf{X})))| \right) \right)$$

$$\leq \hat{\epsilon}_S(h, \hat{h}_S(S)) + 2\mathbb{E} \left[ \sup \frac{1}{N_S} \sum_{i=1}^{N_S} \epsilon_i h(\mathbf{X}_i, y_i, p_i) \right] + C \sqrt{\frac{ln(2/\delta)}{N_S}}$$

$$+ 2C_f C_g W_1 \left( \mathbb{P}_S(G), \mathbb{P}_T(G) \right) + \omega. \tag{40}$$

# F  ALGORITHM

# G  COMPLEXITY ANALYSIS

Here we analyze the computational complexity of the proposed DeSGraDA. The computational complexity primarily relies on Degree-Conscious spiking representations. For a given graph $G$, $\|A\|_0$

---

**Algorithm 1** Learning Algorithm of DeSGraDA

---

**Input:** Source data $\mathcal{D}^s$, Target data $\mathcal{D}^t$
**Output:** The parameters $\theta$ of Degree-Conscious spiking encoder, parameters $\gamma$ of domain discriminator, and parameter $\eta$ of semantic classifier.
 1: Initialize model parameters $\theta$, $\gamma$, and $\eta$
 2: **while** not converged **do**
 3:     Sample mini-batches $\mathcal{B}^s$ and $\mathcal{B}^t$ from $\mathcal{D}^s$ and $\mathcal{D}^t$
 4:     Forward propagate $\mathcal{B}^s$ and $\mathcal{B}^t$ through the Degree-Conscious spiking encoder
 5:     Perform pseudo-label distillation
 6:     Compute the loss function using Eq. 9
 7:     Update model parameters $\theta$, $\gamma$, and $\eta$ via backpropagation
 8: **end while**

---

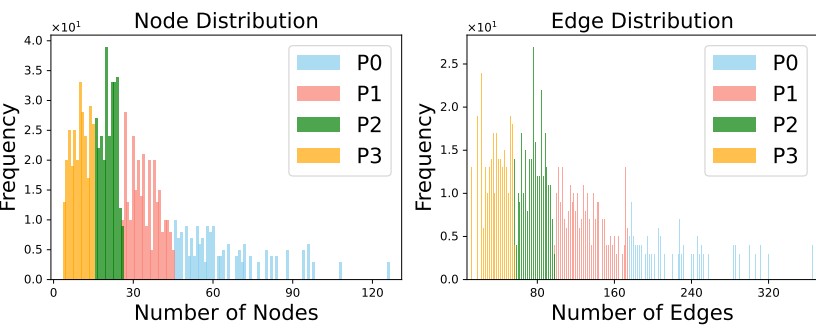

Figure 5: Visualization of different distributions on the PROTEINS dataset.

denotes the number of nonzeros in the adjacency matrix of $G$. $d$ is the feature dimension. $L$ denote the layer number of spiking encoder. $|V|$ is the number of nodes. $T$ denotes the number of latency step. The spiking encoder takes $\mathcal{O}\left(T \cdot L \cdot \left(\|A\|_0 \cdot d + |V| \cdot d^2\right)\right)$ computational time for each graph. As a result, the complexity of our DeSGraDA is proportional to both $|V|$ and $\|A\|_0$.

## H   DATASET

Table 5: Statistics of the experimental datasets.

| Datasets | Graphs | Avg. Nodes | Avg. Edges | Classes |
|----------|--------|------------|------------|---------|
| SEED | 3,818 | 62.00 | 125.74 | 3 |
| BCI | 1,440 | 22.00 | 119.80 | 2 |
| PROTEINS | 1,113 | 39.10 | 72.80 | 2 |
| DD | 1,178 | 284.32 | 715.66 | 2 |
| COX2 | 467 | 41.22 | 43.45 | 2 |
| COX2_MD | 303 | 26.28 | 335.12 | 2 |
| BZR | 405 | 35.75 | 38.36 | 2 |
| BZR_MD | 306 | 21.30 | 225.06 | 2 |

### H.1   DATASET DESCRIPTION

We conduct extensive experiments on different types of datasets. The dataset statistics can be found in Table 5, and their details are shown as follows:

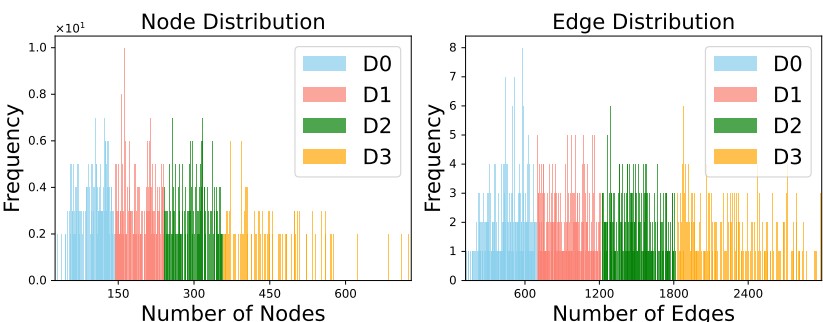

Figure 6: Visualization of different distributions on the DD dataset.

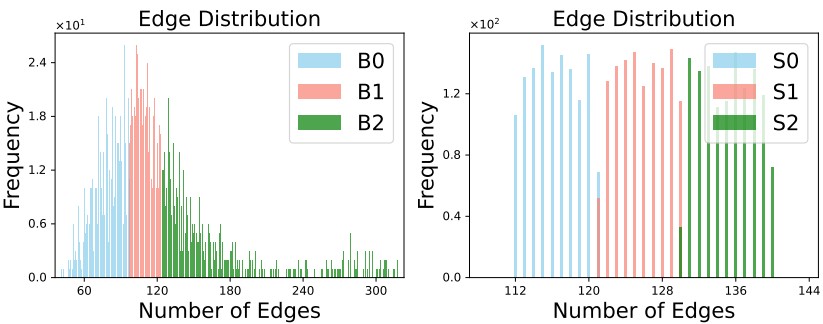

Figure 7: Visualization of edge density distributions on the BCI (left) and SEED (right) datasets.

- **SEED.** The SJTU Emotion EEG Dataset (SEED) (Zheng & Lu, 2015; Duan et al., 2013) is a widely used Electroencephalography (EEG) benchmark for emotion recognition. It contains EEG recordings from 15 participants watching emotional movie clips that evoke positive, neutral, and negative emotions. Each recording consists of 62-channel EEG signals sampled at 1000 Hz. Based on the edge density, we partition the SEED dataset into three sub-datasets, namely S0, S1, and S2. The sub-datasets exhibit substantial domain disparities among them.

- **BCI.** The Brain-Computer Interface (BCI) Competition IV-2a dataset Brunner et al. (2008) is a widely used benchmark for motor imagery EEG classification. It includes EEG recordings from 9 subjects performing four motor imagery tasks: left hand, right hand, feet, and tongue. Each subject completed two sessions on different days, with signals recorded from 22 EEG channels at a sampling rate of 250 Hz. Based on the edge density, we partition the dataset into three parts: B0, B1, and B2.

- **PROTEINS and DD.** The PROTEINS (Dobson & Doig, 2003) and DD datasets comprise protein structure graphs commonly used for graph classification tasks. In both datasets, nodes represent amino acids, forming edges between spatially or sequentially adjacent residues. In PROTEINS, each graph is labeled to indicate whether the protein is an enzyme, with edges defined between residues less than 6 Angstroms apart. The DD dataset, derived from the Protein Data Bank, focuses on classifying proteins by structural class and typically exhibits denser graph connectivity than PROTEINS. Additionally, we partition the PROTEINS and DD datasets into four parts based on edge density and node density, P0 to P3 and D0 to D3, respectively.

- **COX2 and COX2_MD.** The COX2 dataset Sutherland et al. (2003) consists of 467 molecular graphs, while COX2_MD contains 303 structurally modified counterparts. In both datasets, nodes represent atoms and edges correspond to chemical bonds. Specifically, COX2_MD introduces structural variations to the original COX2 molecules while preserving their semantic labels.

- **BZR and BZR_MD.** The BZR dataset Sutherland et al. (2003) comprises 405 molecular graphs, while BZR_MD includes 306 structurally modified graphs derived from BZR. In both datasets, nodes represent atoms and edges denote chemical bonds. BZR_MD introduces structural variations to simulate domain shifts while preserving the original label semantics.

## H.2 DATA PROCESSING

In our implementation, we process the above datasets as follows:

- For datasets from TUDataset [1], including PROTEINS, DD, BZR, BZR_MD, COX2, and COX2_MD, we utilize the TUDataset module from PyTorch Geometric for loading. Self-loops are added during the preprocessing stage to prevent isolated nodes.
- For the SEED dataset, we utilize the TorchEEG library[2] to transform raw EEG signals into graph-structured data. During graph construction, edges are removed following the approach described in Klepl et al. (2022).
- For the BCI dataset, we focus on a binary classification subset involving left-hand and right-hand motor imagery tasks, a widely adopted evaluation setting in BCI research. Following the construction protocols proposed in Altaheri et al. (2022; 2023), we randomly remove a portion of edges from each graph during preprocessing.

## I BASELINES

In this part, we introduce the details of the compared baselines as follows:

**Graph kernel method.** We compare DeSGraDA with one graph kernel method:

- **WL subtree:** Weisfeiler-Lehman (WL) subtree (Shervashidze et al., 2011) is a graph kernel method, which calculates the graph similarity by a kernel function, where it encodes local neighborhood structures into subtree patterns, efficiently capturing the topology information contained in graphs.

**Graph-based methods.** We compare DeSGraDA with four widely used graph-based methods:

- **GCN**: GCN (Kipf & Welling, 2017) is a spectral-based neural network that iteratively updates node representations by aggregating information from neighboring nodes, effectively capturing both local graph structure and node features.
- **GIN**: GIN (Xu et al., 2018) is a message-passing neural network designed to distinguish graph structures using an injective aggregation function, theoretically achieving the expressive power of the Weisfeiler-Lehman test.
- **CIN**: CIN (Bodnar et al., 2021) extends the Weisfeiler-Lehman framework by integrating cellular complexes into graph neural networks, allowing for the capture of higher-dimensional topological features.
- **GMT**: GMT (Baek et al., 2021) utilizes self-attention mechanisms to dynamically adjust the importance of nodes based on their structural dependencies, thereby enhancing both adaptability and performance.

**Spiking-based graph methods.** We compare DeSGraDA with two spiking-based graph methods:

- **SpikeGCN**: SpikeGCN (Zhu et al., 2022) introduces an end-to-end framework designed to integrate the fidelity characteristics of SNNs with graph node representations.
- **DRSGNN**: DRSGNN (Zhao et al., 2024) dynamically adapts to evolving graph structures and relationships through a novel architecture that updates node representations in real-time.

**Domain adaption methods.** We compare DeSGraDA with two recent domain adaption methods:

---

[1] https://chrsmrrs.github.io/datasets/
[2] https://torcheeg.readthedocs.io/en/latest/

Table 6: GPU memory consumption of different graph domain methods in training stage for each training epoch (in GB).

|  | DeSGraDA | DEAL | CoCo | SGDA | StruRW | A2GNN | PA-BOTH |
|---|---|---|---|---|---|---|---|
| PROTEINS | 1.0 | 1.2 | 1.2 | 1.0 | 1.2 | 22.3 | 1.7 |
| DD | 5.6 | 6.4 | 2.5 | 3.9 | 4.5 | 35.1 | 16.8 |
| SEED | 1.1 | 1.4 | 1.5 | 2.6 | 0.8 | 16.8 | 2.8 |
| BCI | 0.7 | 0.8 | 0.7 | 0.7 | 0.7 | 14.5 | 1.6 |

Table 7: Time consumption of different graph domain methods in training stage for each training epoch (in seconds).

|  | DeSGraDA | DEAL | CoCo | SGDA | StruRW | A2GNN | PA-BOTH |
|---|---|---|---|---|---|---|---|
| PROTEINS | 0.195 | 0.103 | 22.123 | 0.088 | 0.088 | 1.313 | 0.949 |
| DD | 0.427 | 0.400 | 184.015 | 0.135 | 0.140 | 2.263 | 0.787 |
| SEED | 0.192 | 0.137 | 26.187 | 0.126 | 0.075 | 1.414 | 0.311 |
| BCI | 0.224 | 0.211 | 35.162 | 0.103 | 0.086 | 0.781 | 0.123 |

- **CDAN**: CDAN (Long et al., 2018) employs a conditional adversarial learning strategy to reduce domain discrepancy by conditioning adversarial adaptation on discriminative information from multiple domains.

- **ToAlign**: ToAlign (Wei et al., 2021b) uses token-level alignment strategies within Transformer architectures to enhance cross-lingual transfer, optimizing the alignment of semantic representations across languages.

- **MetaAlign**: MetaAlign (Wei et al., 2021a) is a meta-learning framework for domain adaptation that dynamically aligns feature distributions across domains by learning domain-invariant representations.

**Graph domain adaptation methods.** We compare DeSGraDA with six graph domain adaption methods:

- **DEAL**: DEAL (Yin et al., 2022) uses domain adversarial learning to align graph representations across different domains without labeled data, overcoming discrepancies between the source and target domains.

- **CoCo**: CoCo (Yin et al., 2023) leverages contrastive learning to align graph representations between source and target domains, enhancing domain adaptation by promoting intra-domain cohesion and inter-domain separation in an unsupervised manner.

- **SGDA**: SGDA (Qiao et al., 2023) utilizes labeled data from the source domain and a limited amount of labeled data from the target domain to learn domain-invariant graph representations.

- **StruRW**: StruRW (Liu et al., 2023) introduces a structural re-weighting mechanism that dynamically adjusts the importance of nodes and edges based on their domain relevance. It enhances feature alignment by emphasizing transferable structures while suppressing domain-specific noise.

- **A2GNN**: A2GNN (Liu et al., 2024a) introduces a novel propagation mechanism to enhance feature transferability across domains, improving the alignment of graph structures and node features in an unsupervised setting.

- **PA-BOTH**: PA-BOTH (Liu et al., 2024b) aligns node pairs between source and target graphs, optimizing feature correspondence at a granular level to improve the transferability of structural and feature information across domains.

## J Implementation Details

DeSGraDA and all baseline models are implemented using PyTorch[3] and PyTorch Geometric[4]. We conduct experiments for DeSGraDA and all baselines on NVIDIA A100 GPUs for a fair comparison, where the learning rate of Adam optimizer set to $10^{-4}$, hidden embedding dimension 256, weight decay $10^{-12}$, and GNN layers 4. Additionally, DeSGraDA and all baseline models are trained using all labeled source samples and evaluated on unlabeled target samples (Wu et al., 2020). The performances of all models are measured and averaged on all samples for five different runs.

## K More experimental results

### K.1 More performance comparison

In this part, we provide additional results for our proposed method DeSGraDA compared with all baseline models across various datasets, as illustrated in Table 13-16. These results consistently show that DeSGraDA outperforms the baselines in most cases, validating the superiority of our proposed method.

Additionally, we find that different domain shift scenarios exhibit similar results in Table 1. However, Table 2 shows that the P0→P2 scenario yields significantly inferior results compared to P0→P1 and P0→P3. To further understand this phenomenon, we analyze the relevant quantitative statistics through the calculation of Wasserstein Distances Panaretos & Zemel (2018) between each pair of sub-datasets. Then, we find that:

- On the SEED and BCI datasets, the adaptation accuracies across the main domain shift scenarios are consistently clustered and exhibit close values. Specifically, for SEED, the accuracies are 58.0%, 57.0%, and 55.9% for S0→S1, S0→S2, and S1→S2, with the corresponding Wasserstein Distances being 0.0052, 0.0047, and 0.0053, respectively. For BCI, the accuracies for B0→B1, B0→B2, and B1→B2 are 54.1%, 56.2%, and 55.0%, with Wasserstein Distances of 0.0044, 0.0053, and 0.0051, respectively. These consistently low values of distributional shift are reflected in the stable adaptation performance observed across the various subdomain pairs in both datasets.

- On the PROTEINS dataset, both node shift and edge shift scenarios demonstrate a strong correspondence between adaptation performance and distributional divergence. For the node domain shift setting, the accuracies for P0→P1, P0→P2, and P0→P3 are 76.3%, 69.2%, and 77.5%, respectively, with Wasserstein Distances of 0.0087, 0.0187, and 0.0081. For the edge domain shift setting, the accuracies are 76.8%, 68.6%, and 76.5% for P0→P1, P0→P2, and P0→P3, with Wasserstein Distances of 0.0089, 0.0254, and 0.0054, respectively. In both settings, the P0→P2 scenario consistently exhibits the lowest performance and the highest distributional divergence, demonstrating that substantial domain shifts, as quantified by larger Wasserstein Distances, lead to pronounced degradation in adaptation performance.

Furthermore, our theoretical analysis in Appendix D also formalizes the connection between domain divergence and transferability using the Wasserstein Distance, thereby providing additional support for our empirical findings.

### K.2 Training time and memory comparison

We provide detailed comparisons of GPU memory consumption and training time per epoch for DeSGraDA and other graph domain adaptation methods under identical experimental settings in this part, as shown in Tables 6 and 7. It is worth noting that the training phase is typically conducted on more powerful hardware to achieve optimal performance within a reasonable time frame.

---

[3]https://pytorch.org/
[4]https://www.pyg.org/

Table 8: The results of DeSGraDA with different widely used graph neural networks (GIN, GCN and SAGE) on the PROTEINS and DD dataset. **Bold** results indicate the best performance.

| Methods | PROTEINS | | | | DD | | | |
|---|---|---|---|---|---|---|---|---|
| | P0→P1 | P1→P0 | P0→P2 | P2→P0 | D0→D1 | D1→D0 | D0→D2 | D2→D0 |
| DeSGraDA w GCN | 71.9 | 76.8 | 64.8 | 68.6 | 58.2 | 70.2 | 57.6 | 64.1 |
| DeSGraDA w SAGE | 73.9 | 79.8 | 65.5 | 74.5 | 57.8 | 71.6 | 59.1 | 66.7 |
| DeSGraDA w GIN | 74.1 | 81.5 | 66.4 | 78.7 | 59.3 | 73.0 | 61.1 | 68.8 |
| DeSGraDA | **76.3** | **84.6** | **69.2** | **83.6** | **60.1** | **76.1** | **63.9** | **71.7** |

Table 9: The results of DeSGraDA with different widely used graph neural networks (GIN, GCN and SAGE) on the SEED and BCI dataset. **Bold** results indicate the best performance.

| Methods | SEED | | | | BCI | | | |
|---|---|---|---|---|---|---|---|---|
| | S0→S1 | S1→S0 | S0→S2 | S2→S0 | B0→B1 | B1→B0 | B0→B2 | B2→B0 |
| DeSGraDA w GCN | 55.7 | 54.5 | 53.7 | 54.0 | 52.8 | 51.6 | 52.2 | 54.2 |
| DeSGraDA w SAGE | 55.3 | 54.5 | 54.2 | 54.9 | 52.7 | 52.2 | 52.8 | 54.7 |
| DeSGraDA w GIN | 56.6 | 57.1 | 55.8 | 56.9 | 53.3 | 52.9 | 53.8 | 55.4 |
| DeSGraDA | **58.0** | **58.2** | **57.0** | **58.3** | **54.1** | **53.6** | **54.9** | **56.2** |

### K.3 MORE ABLATION STUDY

To validate the effectiveness of the different components in DeSGraDA, we conduct more experiments with four variants on DD, SEED, and BCI datasets, i.e., DeSGraDA w CDAN, DeSGraDA w/o PL, DeSGraDA w/o CF and DeSGraDA w/o TL. The results are shown in Table 11 and 12. From the results, we have similar observations as summarized in Section 5.3.

Additionally, we conduct ablation studies to examine the effect of directly replacing the SGNs with commonly used Graph Neural Networks (GNNs) for generating representations for DeSGraDA: (1) DeSGraDA w GCN: It replaces SGNs with GCN (Kipf & Welling, 2017); (2) DeSGraDA w GIN: It replaces SGNs with GIN (Xu et al., 2018); (3) DeSGraDA w SAGE: It replaces SGNs with GraphSAGE (Hamilton et al., 2017). The experimental results across the PROTEINS, DD, SEED, and BCI datasets are shown in Table 8 and 9. However, the critical aspect of our work lies in the specific problem we set up, i.e., low-power and distribution shift environments. In this context, directly replacing SGNs with commonly used GNNs like GIN or GCN is not feasible, as these models are unsuitable for deployment on low-energy devices. As demonstrated in Section 5.4, GNN-based methods have much higher energy consumption than the spike-based methods.

### K.4 MORE SENSITIVITY ANALYSIS

In this part, we provide additional sensitivity analysis of the proposed DeSGraDA with respect to the impact of its hyperparameters: the latency step $T$ and initial threshold value $V_{th}^{degree}$ in SNNs on the DD, SEED and BCI datasets. The results are illustrated in Figure 8 and 9, where we observe trends similar to those discussed in Section 5.5.

Additionally, we conduct a sensitivity analysis of the hyperparameter $\lambda$ in Eq. 9, which balances the adversarial alignment loss, on the PROTEINS dataset. We vary $\lambda$ within the range $\{0.1, 0.3, 0.5, 0.7, 0.9\}$. As shown in Table 10, the results demonstrate that DeSGraDA consistently achieves strong performance across different $\lambda$ values, with the best result obtained at $\lambda = 0.9$. Model performance remains stable for moderate to high values of $\lambda$, indicating that the adversarial alignment loss serves as an effective regularizer without destabilizing the training process.

| | P0→P1 | P1→P0 | P0→P2 | P2→P0 |
|---|---|---|---|---|
| $\lambda = 0.1$ | 75.3 | 84.0 | 68.2 | 82.5 |
| $\lambda = 0.3$ | 75.4 | 83.7 | 68.0 | 82.8 |
| $\lambda = 0.5$ | 75.3 | 84.2 | 68.3 | 82.9 |
| $\lambda = 0.7$ | 75.8 | 84.3 | 68.8 | 82.8 |
| $\lambda = 0.9$ | **76.3** | **84.6** | **69.2** | **83.6** |

Table 10: Hyperparameter sensitivity analysis of $\lambda$ on the PROTEINS dataset.

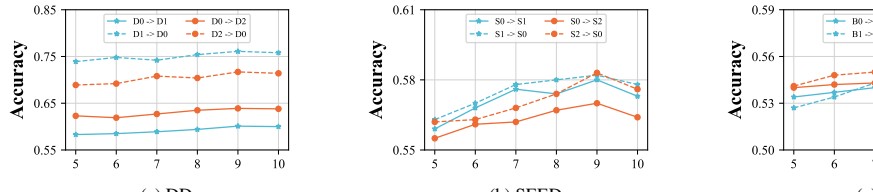

(a) DD        (b) SEED        (c) BCI

Figure 8: Hyperparameter sensitivity analysis of latency step $T$ in SNNs on the DD, SEED and BCI datasets.

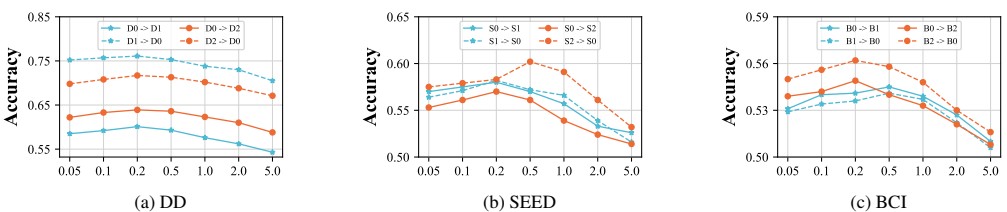

(a) DD        (b) SEED        (c) BCI

Figure 9: Hyperparameter sensitivity analysis of the initial threshold $V_{th}^{degree}$ in SNNs on the DD, SEED and BCI datasets.

## L    LIMITATION

The proposed DeSGraDA framework, while demonstrating significant improvements in spiking graph domain adaptation, does have some limitations. It assumes a substantial domain shift between the source and target domains, which may not always be applicable in real-world scenarios where domain shifts are minimal. Additionally, the computational complexity introduced by adversarial feature distribution alignment and pseudo-label distillation could become a bottleneck, especially for large-scale datasets. The framework's sensitivity to hyperparameters, such as time latency and threshold values, also requires careful tuning for different datasets, which may hinder its practical application. Furthermore, while the method provides a generalization bound, its robustness in diverse real-world settings and its ability to address privacy or fairness concerns in sensitive domains remain underexplored. These aspects highlight opportunities for further refinement and broader applicability of DeSGraDA.

## M    THE USE OF LARGE LANGUAGE MODELS (LLMS)

Large language models (LLMs) were only used to improve the clarity, grammar, and fluency of the manuscript. They were not involved in the development of research ideas, experimental design, data analysis, or any other aspect of the scientific content.

Table 11: The results of ablation studies on the DD dataset (source → target). **Bold** results indicate the best performance.

| Methods | D0→D1 | D1→D0 | D0→D2 | D2→D0 | D0→D3 | D3→D0 | D1→D2 | D2→D1 | D1→D3 | D3→D1 | D2→D3 | D3→D2 |
|---|---|---|---|---|---|---|---|---|---|---|---|---|
| DeSGraDA w CDAN | 57.3 | 72.7 | 59.9 | 68.8 | 57.4 | 58.1 | 68.1 | 60.9 | 75.9 | 59.3 | 80.2 | 77.9 |
| DeSGraDA w/o PL | 58.3 | 71.8 | 58.1 | 69.8 | 57.9 | 58.2 | 67.1 | 59.3 | 75.1 | 60.7 | 79.1 | 76.8 |
| DeSGraDA w/o CF | 51.0 | 62.0 | 57.1 | 61.8 | 54.0 | 51.5 | 59.0 | 54.3 | 62.3 | 52.9 | 64.1 | 69.3 |
| DeSGraDA w/o TL | 58.0 | 72.0 | 60.4 | 68.2 | 58.4 | 58.8 | 68.7 | 60.8 | 77.6 | 60.5 | 76.6 | 77.4 |
| DeSGraDA | **60.1** | **76.1** | **63.9** | **71.7** | **61.1** | **62.3** | **73.6** | **64.2** | **80.9** | **64.7** | **82.0** | **80.6** |

Table 12: The results of ablation studies on the SEED and BCI datasets (source → target). **Bold** results indicate the best performance.

| Methods | SEED | | | | | | BCI | | | | | |
|---|---|---|---|---|---|---|---|---|---|---|---|---|
| | S0→S1 | S1→S0 | S0→S2 | S2→S0 | S1→S2 | S2→S1 | B0→B1 | B1→B0 | B0→B2 | B2→B0 | B1→B2 | B2→B1 |
| DeSGraDA w CDAN | 54.1 | 55.2 | 53.5 | 56.8 | 53.9 | 55.7 | 52.0 | 52.3 | 51.5 | 53.6 | 52.7 | 52.4 |
| DeSGraDA w/o PL | 55.6 | 55.7 | 54.3 | 56.1 | 53.7 | 54.5 | 53.3 | 52.9 | 52.6 | 52.8 | 52.1 | 52.2 |
| DeSGraDA w/o CF | 29.7 | 47.2 | 45.8 | 47.5 | 45.6 | 32.8 | 43.0 | 44.3 | 44.2 | 44.7 | 33.1 | 34.2 |
| DeSGraDA w/o TL | 53.8 | 54.6 | 53.2 | 55.4 | 52.0 | 53.4 | 51.1 | 50.9 | 52.2 | 53.9 | 50.5 | 51.3 |
| DeSGraDA | **58.0** | **58.2** | **57.0** | **58.3** | **55.9** | **58.1** | **54.1** | **53.6** | **54.9** | **56.2** | **55.0** | **54.6** |

Table 13: The graph classification results (in %) on PROTEINS under node density domain shift (source→target). P0, P1, P2, and P3 denote the sub-datasets partitioned with node density. **Bold** results indicate the best performance.

| Methods | P0→P1 | P1→P0 | P0→P2 | P2→P0 | P0→P3 | P3→P0 | P1→P2 | P2→P1 | P1→P3 | P3→P1 | P2→P3 | P3→P2 | Avg. |
|---|---|---|---|---|---|---|---|---|---|---|---|---|---|
| WL subtree | 69.1 | 59.7 | 61.2 | 75.9 | 41.6 | 83.5 | 61.5 | 72.7 | 24.7 | 72.7 | 63.1 | 62.9 | 62.4 |
| GCN | 73.7±0.3 | 82.7±0.4 | 57.6±0.2 | **84.0±1.3** | 24.4±0.4 | 17.3±0.2 | 57.6±0.1 | 70.9±0.7 | 24.4±0.5 | 26.3±0.1 | 37.5±0.2 | 42.5±0.8 | 49.9 |
| GIN | 71.8±2.7 | 70.2±4.7 | 58.5±4.3 | 56.9±4.9 | 74.2±1.7 | 78.2±3.3 | 63.3±2.7 | 67.1±3.8 | 35.9±4.2 | 61.0±2.4 | 71.9±2.1 | 65.1±1.0 | 64.5 |
| GMT | 73.7±0.2 | 82.7±0.1 | 57.6±0.3 | 83.1±0.5 | 75.6±1.4 | 17.3±0.6 | 57.6±1.5 | 73.7±0.6 | 26.3±1.2 | 75.6±0.7 | 42.4±0.5 | 61.8 |
| CIN | 74.1±0.6 | 83.8±1.0 | 60.1±2.1 | 78.6±3.1 | 75.6±0.2 | 74.8±3.7 | 63.9±2.7 | 74.1±0.6 | 57.0±4.3 | 58.9±3.3 | 75.6±0.7 | 63.6±1.0 | 70.0 |
| SpikeGCN | 71.8±0.9 | 80.9±1.4 | 64.9±1.4 | 79.1±2.2 | 71.1±1.9 | 73.8±1.6 | 62.4±2.0 | 71.8±2.3 | 70.1±2.4 | 66.9±1.9 | 72.1±1.9 | 64.5±1.7 | 70.9 |
| DRSGNN | 73.6±1.1 | 81.3±1.5 | 64.6±1.2 | 80.6±1.4 | 70.2±1.7 | 76.1±2.3 | 64.1±1.5 | 71.9±1.9 | 70.4±2.0 | 64.1±3.1 | 74.7±1.4 | 71.3 |
| CDAN | 75.9±1.0 | 83.1±0.6 | 60.8±0.6 | 82.6±0.2 | 75.8±0.3 | 70.9±2.4 | 64.7±0.3 | **77.7±0.6** | 73.3±1.8 | **75.4±0.7** | 75.8±0.4 | 67.1±0.8 | 73.6 |
| ToAlign | 73.7±0.4 | 82.7±0.3 | 57.6±0.6 | 82.7±0.8 | 24.4±0.1 | 82.7±0.3 | 57.6±0.4 | 73.7±0.2 | 24.4±0.7 | 73.7±0.3 | 24.4±0.5 | 57.6±0.4 | 59.6 |
| MetaAlign | 74.3±0.8 | 83.3±2.2 | 60.6±1.7 | 71.2±2.1 | 76.3±0.3 | 77.3±2.4 | 64.6±1.2 | 72.0±1.0 | 76.0±0.5 | 73.3±1.8 | 74.4±1.7 | 56.9±1.4 | 71.7 |
| DEAL | 75.4±1.2 | 78.0±2.4 | 68.1±1.9 | 80.8±2.1 | 73.8±1.4 | 80.6±2.3 | 65.7±1.7 | 74.7±2.4 | 74.7±1.6 | 71.0±2.1 | 68.1±2.6 | 70.3±0.4 | 73.4 |
| CoCo | 74.8±0.6 | 84.1±1.1 | 65.5±0.4 | 83.6±1.1 | 72.4±2.9 | 83.1±0.4 | 69.7±0.5 | 75.8±0.7 | 71.4±2.3 | 73.4±1.3 | 72.5±2.7 | 66.4±1.7 | 74.4 |
| SGDA | 64.2±0.5 | 61.0±0.7 | 66.9±1.2 | 61.9±0.9 | 65.4±1.6 | 66.5±1.0 | 64.6±1.1 | 60.1±0.5 | 66.3±1.3 | 59.3±0.8 | 66.0±1.6 | 66.2±1.3 | 64.1 |
| StruRW | 71.9±2.3 | 82.6±1.9 | 66.7±1.8 | 74.5±2.8 | 52.8±1.9 | 57.3±2.0 | 62.2±2.4 | 63.3±2.1 | 59.5±1.6 | 56.3±2.0 | 66.6±2.3 | 52.4±2.0 | 63.8 |
| A2GNN | 65.7±0.6 | 65.9±0.8 | 66.3±0.9 | 65.6±1.1 | 65.2±1.4 | 65.6±1.3 | 65.9±1.7 | 65.8±1.6 | 65.0±1.5 | 66.1±1.2 | 65.2±1.9 | 65.9±1.8 | 65.7 |
| PA-BOTH | 61.0±0.8 | 61.2±1.3 | 60.3±0.6 | 66.7±2.1 | 63.7±1.5 | 61.9±2.0 | 66.2±1.4 | 69.9±2.3 | 68.0±0.7 | 69.4±1.8 | 61.5±0.4 | 67.6±1.0 | 64.9 |
| DeSGraDA | **76.3±1.9** | **84.6±2.5** | **69.2±2.3** | 83.6±2.6 | **77.5±2.2** | **83.7±1.9** | **69.8±2.4** | 74.0±1.6 | **76.2±2.0** | 73.0±2.1 | **77.8±2.3** | **70.5±1.7** | **76.4** |

Table 14: The graph classification results (in %) on PROTEINS under edge density domain shift (source→target). P0, P1, P2, and P3 denote the sub-datasets partitioned with edge density. **Bold** results indicate the best performance.

| Methods | P0→P1 | P1→P0 | P0→P2 | P2→P0 | P0→P3 | P3→P0 | P1→P2 | P2→P1 | P1→P3 | P3→P1 | P2→P3 | P3→P2 | Avg. |
|---|---|---|---|---|---|---|---|---|---|---|---|---|---|
| WL subtree | 68.7 | 82.3 | 50.7 | 82.3 | 58.1 | 83.8 | 64.0 | 74.1 | 43.7 | 70.5 | 71.3 | 60.1 | 67.5 |
| GCN | 73.4±0.2 | 83.5±0.3 | 57.6±0.2 | 84.2±1.8 | 24.0±0.1 | 16.6±0.4 | 57.6±0.2 | 73.7±0.4 | 24.0±0.1 | 26.6±0.2 | 39.9±0.9 | 42.5±0.1 | 50.3 |
| GIN | 62.5±4.7 | 74.9±3.7 | 53.0±4.6 | 59.6±4.2 | 73.7±0.8 | 64.7±3.4 | 60.6±2.7 | 69.8±0.6 | 31.1±2.8 | 63.1±3.4 | 72.3±2.7 | 64.6±1.4 | 62.5 |
| GMT | 73.4±0.3 | 83.5±0.2 | 57.6±0.1 | 83.5±0.3 | 24.0±0.1 | 83.5±0.1 | 57.4±0.2 | 73.4±0.2 | 24.1±0.1 | 73.4±0.3 | 24.0±0.1 | 57.6±0.2 | 59.6 |
| CIN | 74.5±0.2 | 84.1±0.5 | 57.8±0.2 | 82.7±0.9 | 75.6±0.6 | 79.2±2.2 | 61.5±2.7 | 74.0±1.0 | 75.5±0.8 | 72.5±2.1 | 76.0±0.3 | 60.9±1.2 | 72.9 |
| SpikeGCN | 71.8±0.8 | 79.5±1.3 | 63.8±1.0 | 78.9±1.4 | 68.6±1.1 | 76.5±1.8 | 62.3±2.2 | 72.1±1.5 | 68.1±2.1 | 67.2±1.9 | 69.2±2.1 | 64.2±1.8 | 70.2 |
| DRSGNN | 72.6±0.6 | 80.1±1.6 | 63.1±1.4 | 79.5±1.8 | 70.4±1.9 | 78.6±2.1 | 64.1±1.7 | 70.7±2.3 | 67.8±1.6 | 65.6±1.4 | 71.3±1.3 | 62.1±1.0 | 70.5 |
| CDAN | 72.2±1.8 | 82.4±1.6 | 59.8±2.1 | 76.8±2.4 | 69.3±4.1 | 71.8±3.7 | 64.4±2.5 | 74.3±0.4 | 46.3±2.0 | 69.8±1.8 | 74.4±1.7 | 62.6±2.3 | 68.7 |
| ToAlign | 73.4±0.1 | 83.5±0.2 | 57.6±0.1 | 83.5±0.2 | 24.0±0.3 | 83.5±0.4 | 57.6±0.1 | 73.4±0.1 | 24.0±0.2 | 73.4±0.2 | 24.0±0.1 | 57.6±0.3 | 59.6 |
| MetaAlign | 75.5±0.9 | 84.9±0.6 | 64.8±1.6 | **85.9±1.1** | 69.3±2.7 | 83.3±0.6 | 68.7±1.2 | 74.2±0.7 | 73.3±3.3 | 72.2±0.9 | 69.9±1.8 | 63.6±2.3 | 73.8 |
| DEAL | 76.5±0.4 | 83.1±0.4 | 67.5±1.3 | 77.6±1.8 | 76.0±0.2 | 80.1±2.7 | 66.1±1.3 | 75.4±1.5 | 42.3±4.1 | 68.1±3.7 | 73.1±2.2 | 67.8±1.2 | 71.1 |
| CoCo | 75.5±2.2 | 84.2±0.4 | 59.8±0.5 | 83.4±2.2 | 73.6±2.3 | 81.6±2.4 | 65.8±0.3 | **76.2±0.2** | 75.8±0.2 | 71.1±2.1 | 76.1±2.0 | 67.1±0.6 | 74.2 |
| SGDA | 63.8±0.6 | 65.2±1.3 | 66.7±1.0 | 59.1±1.5 | 60.1±0.8 | 64.4±1.2 | 65.2±0.7 | 63.9±0.9 | 64.5±0.6 | 61.1±1.3 | 58.9±1.4 | 64.9±1.2 | 63.2 |
| StruRW | 72.6±2.2 | 84.5±1.7 | 66.2±2.2 | 72.5±2.4 | 48.9±2.0 | 56.5±2.3 | 63.1±1.8 | 64.4±2.4 | 55.8±2.0 | 56.6±2.4 | 67.0±2.6 | 42.4±2.0 | 62.5 |
| A2GNN | 65.4±1.3 | 66.3±1.1 | 68.2±1.4 | 66.3±1.2 | 65.4±0.7 | 65.9±0.9 | 66.9±1.3 | 65.4±1.2 | 65.6±0.9 | 65.5±1.2 | 66.1±2.0 | 66.0±1.8 | 66.1 |
| PA-BOTH | 63.1±0.7 | 67.2±1.1 | 64.3±0.5 | 72.1±1.6 | 66.3±0.7 | 64.1±1.2 | 69.7±2.1 | 67.5±1.8 | 61.2±1.4 | 67.7±2.1 | 61.2±1.6 | 65.5±0.6 | 65.9 |
| DeSGraDA | **76.8±1.9** | **87.0±2.1** | **68.6±1.8** | 83.7±2.5 | **76.5±2.8** | **83.9±2.3** | **70.3±1.8** | 75.4±2.2 | **76.7±1.9** | **73.7±2.7** | **79.9±3.2** | **67.9±1.3** | **76.7** |

Table 15: The graph classification results (in %) on DD under node density domain shift (source→target). D0, D1, D2, and D3 denote the sub-datasets partitioned with node. **Bold** results indicate the best performance.

| Methods | D0→D1 | D1→D0 | D0→D2 | D2→D0 | D0→D3 | D3→D0 | D1→D2 | D2→D1 | D1→D3 | D3→D1 | D2→D3 | D3→D2 | Avg. |
|---|---|---|---|---|---|---|---|---|---|---|---|---|---|
| WL subtree | 49.2 | 56.8 | 29.6 | 20.1 | 21.0 | 18.4 | 59.5 | 50.5 | 57.3 | 48.1 | 63.9 | 66.9 | 49.3 |
| GCN | 48.9±2.8 | 59.0±1.7 | 20.7±2.0 | 27.3±2.3 | 15.1±1.8 | 26.9±2.2 | 61.6±1.9 | 53.6±1.5 | 68.1±1.6 | 52.9±2.6 | 64.9±2.1 | 69.7±2.3 | 47.4 |
| GIN | 48.8±1.9 | 24.7±2.1 | 44.1±1.8 | 22.4±2.3 | 57.0±2.1 | 18.4±2.0 | 73.0±1.8 | 52.5±2.3 | 63.2±1.6 | 53.6±1.5 | 70.3±2.6 | 69.4±1.8 | 49.6 |
| GMT | 49.1±1.9 | 32.9±2.2 | 31.8±1.8 | 27.3±2.3 | 52.5±1.5 | 27.6±1.8 | 75.4±1.9 | 53.2±2.1 | 74.1±2.6 | 57.9±2.4 | 70.9±1.8 | 71.1±2.7 | 52.0 |
| CIN | 50.4±1.8 | 18.4±2.0 | 21.2±2.1 | 36.8±1.8 | 43.0±2.1 | 22.9±1.9 | 53.4±1.7 | 56.5±1.5 | 62.3±1.6 | 53.3±1.9 | 75.0±2.1 | 69.3±2.0 | 51.4 |
| SpikeGCN | 53.4±1.3 | 61.5±1.1 | 28.4±2.1 | 48.0±1.4 | 20.8±1.4 | 51.2±1.4 | 69.3±2.0 | 62.4±1.6 | 77.9±1.1 | 64.1±1.5 | 76.8±2.1 | 71.9±1.8 | 57.1 |
| DRSGNN | 52.1±1.5 | 69.7±1.8 | 28.7±1.7 | 42.4±2.1 | 18.6±2.0 | 48.3±2.8 | 79.5±2.0 | 59.5±1.5 | 77.7±1.9 | 63.1±1.5 | 76.4±2.1 | 72.1±2.3 | 57.4 |
| CDAN | 49.6±2.2 | 69.8±1.6 | 44.1±1.8 | 33.9±1.9 | 43.1±2.4 | 42.3±2.0 | 70.5±1.8 | 60.3±2.2 | 76.6±2.1 | 60.1±1.4 | 75.8±2.5 | 70.5±2.3 | 58.1 |
| ToAlign | 54.0±2.4 | 71.0±2.1 | 34.8±1.8 | 46.9±1.9 | 29.6±2.3 | 45.7±1.8 | 71.9±2.2 | 61.6±2.1 | 76.7±1.9 | 62.8±2.3 | 76.4±2.0 | 71.0±1.3 | 58.5 |
| MetaAlign | 48.1±2.0 | 70.0±1.9 | 30.7±1.4 | 18.4±1.8 | 24.9±2.3 | 18.4±1.9 | 70.1±2.3 | 51.9±1.5 | 74.6±2.4 | 51.9±2.2 | 75.1±1.8 | 69.3±1.7 | 52.7 |
| DEAL | 57.9±2.3 | 71.6±2.2 | 57.2±1.8 | 59.3±1.5 | **62.2±2.2** | 59.4±1.9 | 72.1±2.3 | 63.9±1.7 | 78.2±2.2 | 62.6±1.8 | 78.3±2.1 | 77.3±1.9 | 66.5 |
| CoCo | 59.5±2.1 | 70.4±1.7 | 56.6±2.6 | 58.3±2.2 | 59.4±1.9 | 53.9±2.5 | **74.7±1.4** | 62.7±2.0 | 70.6±1.6 | 63.1±2.0 | 77.2±1.7 | 76.4±2.4 | 65.2 |
| SGDA | 57.7±1.5 | 63.8±2.1 | 49.8±2.3 | 54.1±1.7 | 42.6±2.3 | 54.9±1.6 | 68.9±1.5 | 63.0±2.3 | 78.7±2.7 | 64.5±2.1 | 75.9±1.4 | 74.2±1.6 | 62.9 |
| StruRW | 50.0±2.3 | 53.1±1.9 | 32.4±2.4 | 40.6±2.1 | 26.0±2.4 | 38.4±2.0 | 73.3±1.6 | 61.7±1.8 | 71.2±1.9 | 53.6±2.3 | 75.2±2.1 | 71.0±2.2 | 53.9 |
| A2GNN | 56.1±2.0 | 68.5±1.6 | 48.7±2.1 | 52.5±1.8 | 42.9±1.4 | 48.4±1.8 | 70.8±1.7 | 51.9±2.0 | 76.3±2.2 | 51.9±1.8 | 75.1±1.6 | 69.3±2.4 | 59.4 |
| PA-BOTH | 51.4±1.8 | 62.7±2.1 | 31.8±2.0 | 40.5±2.3 | 28.5±1.9 | 45.0±2.2 | 69.5±1.6 | 61.0±1.5 | 68.4±1.7 | 57.8±2.2 | 73.0±2.4 | 73.3±2.5 | 55.3 |
| DeSGraDA | **60.1±2.2** | **76.1±1.8** | **63.9±1.9** | **71.7±2.0** | 61.1±1.9 | **62.3±1.6** | 73.6±2.3 | **64.2±2.0** | **80.9±2.2** | **64.7±1.8** | **82.0±2.5** | **80.6±2.1** | **70.1** |

Table 16: The graph classification results (in %) on DD under edge density domain shift (source→target). D0, D1, D2, and D3 denote the sub-datasets partitioned with node. **Bold** results indicate the best performance.

| Methods | D0→D1 | D1→D0 | D0→D2 | D2→D0 | D0→D3 | D3→D0 | D1→D2 | D2→D1 | D1→D3 | D3→D1 | D2→D3 | D3→D2 | Avg. |
|---|---|---|---|---|---|---|---|---|---|---|---|---|---|
| WL subtree | 51.5 | 59.5 | 28.6 | 23.8 | 23.4 | 19.4 | 56.8 | 51.2 | 54.9 | 50.2 | 61.5 | 57.3 | 44.8 |
| GCN | 49.6±2.2 | 62.7±2.3 | 22.8±2.0 | 26.9±1.4 | 13.9±2.0 | 22.6±1.9 | 74.6±1.3 | 58.7±2.4 | 75.1±1.1 | 52.2±1.6 | 76.6±1.3 | 67.5±2.1 | 50.3 |
| GIN | 48.9±2.8 | 25.9±1.8 | 44.6±1.5 | 23.0±2.0 | 57.2±1.8 | 19.4±2.0 | 71.8±1.2 | 54.8±2.4 | 62.2±1.5 | 52.7±1.9 | 71.3±1.7 | 67.5±2.4 | 50.0 |
| GMT | 50.8±2.2 | 42.7±2.5 | 34.9±2.7 | 48.2±2.0 | 29.6±2.5 | 24.6±1.6 | 68.9±1.9 | 52.6±1.6 | 71.2±1.5 | 52.7±1.3 | 75.9±1.4 | 67.8±1.6 | 52.9 |
| CIN | 50.4±2.2 | 29.4±2.0 | 23.1±1.4 | 31.6±1.7 | 42.8±2.0 | 24.6±1.6 | 54.2±1.2 | 57.5±1.6 | 73.5±2.2 | 52.7±1.3 | 75.6±1.6 | 67.1±2.1 | 48.6 |
| SpikeGCN | 56.4±1.9 | 70.5±2.1 | 34.1±2.6 | 53.2±2.9 | 20.7±1.6 | 49.1±1.7 | 79.7±2.4 | 66.5±1.2 | 77.3±2.1 | 61.7±1.6 | 78.7±2.0 | 71.0±1.5 | 59.9 |
| DRSGNN | 55.3±2.4 | 69.9±2.2 | 27.4±2.0 | 47.6±2.7 | 17.9±1.6 | 47.4±2.1 | 70.7±2.0 | 65.9±1.7 | 76.9±2.1 | 62.2±1.4 | 78.5±1.8 | 71.4±1.6 | 57.6 |
| CDAN | 49.7±1.9 | 65.3±2.3 | 45.4±1.8 | 43.1±2.1 | 42.8±1.7 | 51.8±1.6 | 71.5±2.0 | 64.9±1.6 | 74.5±2.5 | 59.2±2.2 | 77.9±2.1 | 69.0±1.5 | 59.5 |
| ToAlign | 52.3±2.5 | 66.5±2.0 | 47.1±1.6 | 45.6±1.8 | 41.2±2.2 | 51.2±1.8 | 73.9±1.9 | 65.9±2.3 | 77.6±2.0 | 60.8±1.6 | 78.1±2.4 | 70.2±2.1 | 60.9 |
| MetaAlign | 48.1±2.0 | 67.3±1.7 | 32.8±2.0 | 19.4±1.8 | 23.9±2.5 | 19.4±1.7 | 70.1±1.8 | 51.9±2.1 | 77.3±3.2 | 51.9±1.6 | 76.1±1.8 | 70.5±2.0 | 50.7 |
| DEAL | 58.4±1.5 | 70.6±2.0 | 63.9±1.6 | 54.1±2.1 | 66.9±2.4 | 51.8±1.6 | 75.1±2.5 | **67.4±1.8** | 77.8±1.9 | 60.3±2.1 | 80.5±1.4 | 75.0±2.0 | 66.8 |
| CoCo | 60.9±2.3 | 69.6±1.2 | 62.2±2.2 | 66.2±2.0 | 66.0±1.8 | 52.5±2.3 | 71.1±2.4 | 65.3±1.5 | 78.9±1.3 | 60.3±1.4 | 79.6±2.1 | 73.5±1.8 | 67.3 |
| SGDA | 57.2±1.5 | 68.8±1.8 | 42.3±2.0 | 61.4±1.7 | 39.8±2.2 | 52.0±1.6 | 66.7±1.9 | 66.4±2.3 | 78.1±2.1 | 63.6±2.6 | 73.6±1.6 | 70.8±1.9 | 61.7 |
| StruRW | 52.5±2.5 | 56.7±1.3 | 39.0±2.3 | 40.1±2.0 | 24.4±2.1 | 29.7±2.4 | 75.4±1.7 | 63.3±2.0 | 74.8±1.6 | 53.4±1.5 | 75.4±1.4 | 68.7±1.7 | 54.6 |
| A2GNN | 53.1±2.0 | 65.3±1.7 | 42.8±1.9 | 40.5±2.1 | 33.9±2.5 | 39.4±1.8 | 69.8±2.2 | 61.9±1.9 | 77.3±2.1 | 61.9±2.0 | 76.1±2.3 | 67.2±1.8 | 57.4 |
| PA-BOTH | 51.9±1.8 | 50.6±2.0 | 35.8±1.5 | 37.7±1.7 | 27.6±2.3 | 43.7±1.9 | 62.1±1.6 | 61.2±1.9 | 65.7±2.0 | 58.2±1.5 | 73.3±1.8 | 69.5±2.5 | 53.1 |
| DeSGraDA | **62.1±2.0** | **72.0±2.4** | **69.1±1.7** | **71.7±2.2** | **68.7±1.6** | **58.6±2.1** | **76.1±1.9** | 66.4±1.7 | **80.7±2.2** | **63.6±1.6** | **81.9±1.5** | **78.6±2.3** | **70.8** |

