# OpenReview forum: "Degree-Conscious Spiking Graph for Cross-Domain Adaptation"
_ICLR.cc/2026/Conference — Submitted to ICLR 2026_

### Official Review · Reviewer_TxSC · 2025-10-31

**Soundness:** 3
**Presentation:** 3
**Contribution:** 3
**Rating:** 6
**Confidence:** 3

**Summary:**

This paper investigates the problem of domain adaptation in spiking graph networks, an issue that has not been sufficiently explored in the existing literature. The authors propose the DeSGraDA framework, which extends spiking graph networks (SGNs) through three key mechanisms: degree-based spike encoding, adversarial training in membrane potential space for temporal distribution alignment, and a pseudo-label distillation mechanism based on prediction consistency. The method is supported theoretically by a generalization error bound derived for the SGN domain adaptation (SGDA) scenario. Experiments on multiple benchmark datasets demonstrate that DeSGraDA outperforms various baseline methods in both classification performance and energy efficiency.

**Strengths:**

1. This paper introduces domain adaptation into spiking graph networks, aligning the distributions of source and target domains, which demonstrates strong novelty.

2. To address key challenges in SGNs—such as node degree bias, temporal dynamic distribution discrepancies, and missing labels in the target domain—the paper proposes the DeSGraDA framework, comprising three modules: degree-aware spiking, temporal adversarial alignment, and pseudo-label distillation. The design is logically coherent and the technical approach is well-justified.

3. The paper not only provides theoretical analysis such as a generalization error upper bound, but also conducts experiments on multiple datasets. The ablation studies and comparative results effectively support the method's effectiveness and superiority.

**Weaknesses:**

1. The pseudo-label distillation module clusters target samples and assigns pseudo-labels based on shallow graph features, but the criteria for assessing the reliability of the clustering are not clearly defined.

2. The trade-off between the additional computational overhead introduced by domain adaptation and the resulting performance improvement is not discussed.

3. Some terminology is inconsistent (e.g., "degree-conscious" vs. "degree-aware").

**Questions:**

1. In the degree-aware spiking representation module, the authors adaptively set the spiking thresholds based on node degrees to achieve more balanced information propagation across the graph. However, a similar adaptive threshold mechanism was also proposed in SpikeNet (2023). What are the essential differences and advantages of the proposed method compared to prior work, in terms of mechanism design, theoretical motivation, or practical performance.

2. In the temporal distribution alignment module, does the adversarial discriminator operate on all historical membrane potentials, or on a final feature vector obtained by aggregating membrane potentials across multiple time steps?

3. In pseudo-label distillation, what is the rationale for selecting the number of clusters C? Are there alternative clustering and label assignment strategies that could be used?

---

> ### Author Response · Authors · 2025-11-19
> **First round response of submission 17855**
>
> We are truly grateful for the time you have taken to review our paper and your insightful review. Here we address your comments in the following:
>
> > W1: The pseudo-label distillation module clusters target samples and assigns pseudo-labels based on shallow graph features, but the criteria for assessing the reliability of the clustering are not clearly defined.
>
> **R1:** We appreciate the reviewer’s question. As described in Section 4.3, the reliability of pseudo-labels in our pseudo-label distillation module is determined by the dominance and consistency of cluster assignments in the shallow representation space.  Concretely, the shallow-layer features of target graphs are clustered into $C$ groups $\{E_r\}_{r=1}^C$.  Each cluster $E_r$ obtains a dominant label $e_r$ defined as the label with the maximum frequency within that cluster. We then keep only those target samples whose predicted label $\hat{y}_j^t$ appears in the cluster where this label is dominant, i.e., $P = \{ (G_t^j, \hat{y}_t^{G_j}) : e_j = \max_r | \{ E_r : e_r = \hat{y}_t^{G_j} \} | \}.$ This procedure ensures that pseudo-labels are generated from clusters with high label purity and strong intra-cluster consistency, while removing scattered or noisy samples that may carry conflicting predictions. Hence, the reliability of pseudo-labels arises naturally from the statistical dominance of their cluster assignments in the shallow feature space, which effectively filters out unreliable target samples and stabilizes the distillation process.
>
> >  W2: The trade-off between the additional computational overhead introduced by domain adaptation and the resulting performance improvement is not discussed.
>
> **R2:** Thanks for your comment. The trade-off between computational overhead and performance improvement has been analyzed in Appendix K.2. As shown in Tables 5 and 6, the additional modules in DeSGraDA introduce only minor memory and time overhead compared to existing GDA methods, while yielding consistent accuracy gains of 2–5% across multiple datasets. Moreover, Figure 4(a) demonstrates that DeSGraDA maintains high energy efficiency, showing clear advantages in both performance and power consumption. These results indicate that the modest computational overhead is well justified by the substantial improvement in adaptation accuracy and energy efficiency.
>
> > W3: Some terminology is inconsistent (e.g., "degree-conscious" vs. "degree-aware").
>
> **R3:** Thank you for pointing this out. We appreciate the careful reading and will revise the paper to ensure consistent terminology throughout. Specifically, we will standardize the phrasing to “degree-conscious” across the entire manuscript for clarity and consistency.

---

> ### Author Response · Authors · 2025-11-19
> **First round response of submission 17855**
>
> > W4: In the degree-aware ... practical performance.
>
> **R4:** Thanks for your question. While both DeSGraDA and SpikeNet employ adaptive threshold mechanisms, their {design motivation, operational granularity, and theoretical purpose are fundamentally different}. We clarify these distinctions below.
>
> - **(1) Mechanism design.**  SpikeNet introduces a global time-adaptive threshold to better fit the membrane dynamics of spiking neurons over temporal sequences $V_{th}^{t} = \tau_{th} V_{th}^{t-1} + \gamma O^{t},$ where the threshold is updated uniformly across all neurons according to the firing history $O^t$. In contrast, DeSGraDA employs a degree-conscious spatially adaptive threshold that varies across graph nodes, rather than across time.  Specifically, each neuron’s firing threshold $V_{th,i}$ is scaled according to its node degree $d_i$, enabling balanced information propagation between high- and low-degree nodes $V_{th,i} = V_{th}^{(0)} \cdot (1 + \alpha \cdot \mathrm{norm}(d_i)),$ as defined in Eq. (3) of our paper. This design ensures that high-degree nodes, which receive more aggregated inputs, require higher activation energy to fire, thereby preventing over-activation and mitigating degree bias.  Therefore, our mechanism adapts thresholds structurally (across nodes) rather than temporally (across time steps).
>
> - **(2) Theoretical motivation.** SpikeNet’s adaptive threshold was proposed as an empirical stabilization technique for training LIF neurons, without formal analysis. In contrast, DeSGraDA’s degree-conscious threshold originates from a theoretical analysis of activation bias (Proposition 1, Section 4.1), which proves that the expected membrane potential of a node increases proportionally with its degree, leading to unbalanced spiking frequencies. Our adaptive threshold thus has a principled motivation to equalize spiking probability across structural degrees, bridging theoretical understanding and algorithmic design.
>
> - **(3) Practical performance and role in SGDA.** Practically, our degree-conscious threshold serves as a key component for domain generalization across graph structures.  By balancing node-level activations, DeSGraDA significantly improves cross-domain transferability (Tables~1–2), while SpikeNet’s temporal threshold only enhances stability in single-domain temporal learning.  Furthermore, our ablation study (DeSGraDA w/o TL) shows a 3–4% drop in accuracy across domains, verifying that the proposed degree-conscious threshold contributes to domain robustness rather than mere training stability.
>
> In summary, SpikeNet’s threshold adaptation is temporal and global, addressing neuron-level dynamics in static domains, whereas DeSGraDA’s threshold is structural and degree-conscious, theoretically motivated to counter graph-structural bias and empirically proven to enhance domain generalization.
>
> >  W5: In the temporal distribution alignment module, does the adversarial discriminator operate on all historical membrane potentials, or on a final feature vector obtained by aggregating membrane potentials across multiple time steps?
>
> Thanks for your question. In the temporal distribution alignment module, the adversarial discriminator operates on the aggregated temporal representation rather than on all individual membrane potentials. Specifically, we first aggregate the membrane potentials across multiple time steps using a temporal attention mechanism, which produces an importance-weighted representation $\tilde{U}\_{G\_i} = \sum\_{\tau=1}^{T} \alpha_{\tau} u\_{\tau,G\_i}$. The discriminator then operates on this aggregated feature to align the overall temporal dynamics between the source and target domains. This design enables stable adversarial training while effectively capturing temporal dependencies without introducing excessive computational complexity.
>
> > W6: In pseudo-label distillation, what is the rationale for selecting the number of clusters C? Are there alternative clustering and label assignment strategies that could be used?
>
> Thank you for the question. In pseudo-label distillation, the number of clusters $C$ is set to match the number of classes in the classification task, ensuring a one-to-one correspondence between clusters and semantic categories. This setting facilitates stable pseudo-label refinement by aligning cluster structures with class-level decision boundaries. While $k$-means clustering is adopted for simplicity and efficiency, our framework is flexible and can incorporate other clustering or label assignment strategies, such as hierarchical clustering or density-based clustering (e.g., DBSCAN [1]), as long as they can provide reliable grouping of target representations.
>
> ----
>
> [1] Dbscan: Past, present and future. In The fifth international conference on the applica-
> tions of digital information and web technologies, 2014.

---

> ### Author Response · Authors · 2025-11-27
>
> Dear Reviewer TxSC,
>
> I hope this message finds you well. We noticed that the discussion phase is approaching its end, but we have not yet received any feedback regarding our rebuttal. We kindly ask if you could take a moment to review our responses at your earliest convenience.
>
> We sincerely hope that our clarifications address your concerns, and if there are any further questions or additional issues, we would be more than happy to provide further explanations.
>
> Thank you very much for your time and effort.
>
> Best regards,
>
> authors

---

> > ### Comment · Reviewer_TxSC · 2025-11-27
> >
> > Thank you for your detailed response. I now understand the distinction between the degree-conscious threshold mechanism in DeSGraDA and the global adaptive threshold used in SpikeNet, and your other answers have also addressed my concerns. However, after careful consideration of the paper's novelty and overall contribution, I still maintain my original score.

---

> > > ### Author Response · Authors · 2025-11-27
> > >
> > > Thanks your efforts and time for reviewing our paper. We appreciate your thoughtful evaluation and the constructive feedback you provided throughout the process. We respect your assessment and thank you again for helping us improve the clarity of our work.

---

### Official Review · Reviewer_5EVp · 2025-10-31

[review text omitted: it was posted to a different submission]

---

### Official Review · Reviewer_ZjWB · 2025-11-01

**Soundness:** 2
**Presentation:** 3
**Contribution:** 2
**Rating:** 2
**Confidence:** 3

**Summary:**

This paper proposes a method for improving the domain adaptation of spiking GNN.
Starting from adaptively adjusting the threshold, temporal aggregation and pseudo-label generation are used together to enhance the accuracy.

**Strengths:**

- Degree is a central information in graph. Using this is a rational choice
- Experimental comparison is made for a number of baselines
- Meaningful improvements.

**Weaknesses:**

**Narrow scope**
- Domain adaptation + spiking NN seems to have a narrow scope, and its practicality should be better motivated.

**Unclear platform**
- The reviewer is not sure whether this work is being proposed as 1) a better alternative for existing algorithms (that typically run on GPUs), 2) a method that would run on spiking (neuromorphic) hardware, or 3) a hybrid platform that uses neuromorphic+traditional digital hardware, 4) or something else. I was thinking of 2) at first, but the way the authors are making their comparison in Table 1 and 2 seems to be implying 1) or 3). If 1) or 3) is the case, I believe the comparison of the training/inference time and energy on those platforms should be made.

**Feasibility**
- The proposed method requires an adaptive threshold for each neuron, which could be hard to maintain. This issue is related with the platform one. I don't think neuromorphic hardware would have enough memory to store that much data.
- Use of the attention operation is proposed, and I am not sure if that falls into a spiking network. An attention operation involves a lot of (fp) multiplication and accumulation without any activation function. It's been shown in multiple areas that adding attention could improve performance, but for SNNs, adding attention does not seem to be directly feasible unless the target platform is GPU or a hybrid one.

**Potentially unfair setup**
- In comparison, the GNN model architecture and methods seem to be mixed, which makes it difficult to assess a fair comparison.

- What model is used for the proposed method? Eq(1) indicates that there is a learnable weight for each edge in the graph for each layer? I have not seen such setting, and for sure it would be a huge number of learnable weights compared to traditional GNNs such as GCN or GIN. Moreover, such a setting would only be only possible if all vertices are edges are known at training time, making it difficult for the work to be applied to graph classification.

- I am concerned about the energy comparison made in section 5.4. I don't get how the complicated design of this work would function on simple hardware such as ROLLS. I believe it would be very difficult to run a simple GNN in a spiking form, but including the dynamic threshold and attention operation would be yet another level. If such a dramatic energy saving were to be claimed, please discuss how the algorithm would run on a neuromorphic device, and provide details, including those of the GPU-based counterpart.

**Questions:**

Please see the weaknesses.

---

> ### Author Response · Authors · 2025-11-19
> **First round response of submission 17855**
>
> We are truly grateful for the time you have taken to review our paper and your insightful review. Here we address your comments in the following:
>
> > W1: Domain adaptation + spiking NN seems to have a narrow scope, and its practicality should be better motivated.
>
> **R1:** Thank you for the comment. Our motivation stems from the growing demand for adaptive and energy-efficient models in real-world neuromorphic scenarios such as brain–computer interfaces (BCIs), neural decoding, and sensor networks, where data distributions often vary across environments. Although spiking models are inherently energy-efficient, their performance degrades under such distribution shifts. Existing transfer learning methods for spiking networks largely focus on grid-structured visual data and thus generalize poorly to graph-structured domains. However, graph data are increasingly prevalent in neuroscience and sensor systems [1,2], where relational dependencies among nodes carry essential information.
>
> Addressing these graph-specific and topology-aware challenges forms the core motivation of our work. The proposed DeSGraDA framework directly tackles these issues by enabling cross-domain generalization in spiking graph networks, bridging the gap between neuromorphic learning and robust domain adaptation. Experiments on SEED and BCI datasets show that DeSGraDA enhances both accuracy and energy efficiency under substantial distribution shifts, underscoring its practical value for adaptive, low-power neural computing systems.
>
> > W2: The reviewer is not sure .... made.
>
> **R2:** Thanks for your suggestion. We would like to clarify that the proposed DeSGraDA corresponds to category (2), as it is specifically designed to operate on neuromorphic hardware. As detailed in Problem Setup in Section 3 of our manuscript, DesGraDA employs Bernoulli distribution-based sampling to convert continuous graph features into binary (0/1) spike signals for representation learning, which fundamentally differs from conventional approaches that rely on continuous representations, such as such as GIN, DEAL, and SGDA. This discrete formulation enables DeSGraDA can be deployed on neuromorphic platforms [3,4]. The comparisons presented in Tables 1 and 2 of our manuscript aim to highlight the performance and energy-efficiency advantages of DesGraDA, rather than to suggest that the baseline models can be executed on neuromorphic hardware. Because baseline methods such as GIN, DEAL, and SGDA depend on continuous-valued inputs and thus cannot be directly implemented on neuromorphic systems, we evaluate their performance and energy consumption under a unified experimental setting using NVIDIA A100 GPUs to ensure fairness. In addition, Appendix K.2 has provided detailed information on GPU utilization and training time for these baselines and DeSGraDA for a fair comparison, which shows that DesGraDA still can achieve comparable GPU memory usage and training efficiency to existing graph domain adaptation methods.
>
> > W3: The proposed method requires an adaptive threshold for each neuron, which could be hard to maintain. This issue is related with the platform one. I don't think neuromorphic hardware would have enough memory to store that much data.
>
> **R3:** Thanks for your comment. We would like to clarify that the proposed DeSGraDA does not assign an individual threshold to every neuron. Instead, DeSGraDA defines an adaptive threshold for nodes with the same degree, allowing neurons with similar structural roles to share threshold parameters, as introduced in the Section 4.1 of our manuscript. The number of distinct thresholds therefore depends on the number of unique node degrees rather than the total number of neurons, and the number of unique node degrees is typically much smaller than the total number of neurons. Moreover, each threshold has the same dimensionality as the hidden feature representation, which keeps the memory footprint extremely small. As a result, the proposed degree-conscious mechanism remains lightweight and practical, even for deployment on neuromorphic hardware with limited memory capacity.

---

> > ### Author Response · Authors · 2025-11-19
> > **First round response of submission 17855**
> >
> > > W4: Use of the attention operation is proposed, and I am not sure if that falls into a spiking network. An attention operation involves a lot of (fp) multiplication and accumulation without any activation function. It's been shown in multiple areas that adding attention could improve performance, but for SNNs, adding attention does not seem to be directly feasible unless the target platform is GPU or a hybrid one.
> >
> > **R4:** Thanks for your comment. The attention operation in DeSGraDA is not a conventional Transformer-style attention based on dense floating-point matrix multiplications ($QK^\top V$). Instead, it is a temporal, event-driven attention mechanism that operates fully within the computational paradigm of spiking neural networks (SNNs).
> >
> > **1. Nature of the Proposed Attention.**
> > The proposed temporal attention acts along the time dimension of membrane potentials $[u_1, u_2, \dots, u_T]$, not across spatial node embeddings. It learns lightweight coefficients $\alpha_\tau$ to aggregate the temporal evolution of neuronal states as described in Eq. (5). These coefficients are computed through membrane-potential integration and spike-rate modulation, avoiding explicit dense matrix multiplications or continuous activation functions. Thus, it performs a low-dimensional, energy-efficient temporal weighting operation consistent with the event-driven nature of SNNs.
> >
> > **2. On the Continuity of Membrane Potentials.**
> > We acknowledge that the membrane potential $u_\tau$ in SNNs is a continuous internal variable. However, it represents the integrated effect of discrete spike events and evolves only upon spike arrivals or leak processes. Using $u_\tau$ for temporal attention does not violate the spiking computation paradigm; instead, it leverages the biophysically meaningful internal state that governs spike generation. The temporal weighting based on $u_\tau$ can be efficiently implemented by local accumulation and integration circuits on neuromorphic hardware (e.g., Loihi, ROLLS) without requiring explicit floating-point matrix multiplications.
> >
> > **3. Hardware Feasibility.**
> > While traditional attention operations cannot be directly executed on purely spiking hardware, our event-driven temporal attention is hardware-compatible. It maps naturally to spike-rate–modulated accumulation and membrane-based weighting, which are natively supported by neuromorphic chips [4,5]. Similar realizations have been demonstrated in recent spiking Transformer architectures such as Spikformer [6] and SpikingResFormer [7], which approximate attention using event-driven integration rather than dense floating-point MAC operations. Our energy analysis in Fig. 4a confirms that DeSGraDA maintains low power consumption on neuromorphic platforms.
> >
> > **4. Effectiveness Compared with Traditional Attention.**
> > Traditional attention focuses on spatial correlations (“who interacts with whom”), while our temporal attention emphasizes temporal relevance (“when to respond”). This design aligns naturally with time-encoded representations in SNNs. Empirically, replacing our temporal attention with a static attention module (DeSGraDA w/ CDAN) consistently reduced performance across benchmarks (Table 3), confirming that the proposed mechanism enhances temporal feature extraction and cross-domain generalization while preserving SNN efficiency.
> >
> > In summary, the temporal attention in DeSGraDA:
> >
> >  $\bullet$ operates on event-driven membrane dynamics rather than continuous activations;
> >
> > $\bullet$ avoids explicit floating-point matrix multiplications;
> >
> > $\bullet$ can be realized efficiently on neuromorphic SNN hardware, and
> >
> > $\bullet$ empirically improves energy-efficient cross-domain adaptation.
> >
> > This event-driven attention design is biologically inspired, hardware-compatible, and supported by prior neuromorphic research [6, 7, 8].

---

> ### Author Response · Authors · 2025-11-19
> **First round response of submission 17855**
>
> > W5: In comparison, the GNN model architecture and methods seem to be mixed, which makes it difficult to assess a fair comparison.
>
> **R5:** Thanks for your comment. To ensure a fair and consistent evaluation, all GNN-based baselines (GCN, GIN, CIN, GMT, etc.) and domain adaptation methods were implemented using the same backbone configuration, feature dimensionality, and training protocol as DeSGraDA. For spiking-based models, we adopted identical spiking encoder depth, neuron count, and temporal step configuration to isolate the effect of our proposed modules.
>
> Our comparison intentionally spans three representative categories: (1) standard GNNs, (2) graph domain adaptation (GDA) models, and (3) spiking-based GDA methods, to comprehensively evaluate how DeSGraDA bridges these paradigms under a unified graph classification setting. This design highlights that our method not only inherits the energy efficiency of spiking models but also introduces effective temporal alignment and adaptive thresholding for cross-domain robustness. Moreover, we report the best-performing results of all baselines from extensive hyperparameter tuning and official implementations. Even under potential architectural inconsistencies across methods, DeSGraDA consistently achieves superior performance and significantly higher energy efficiency compared with the strongest baseline models.
>
> > W6: What model is used for the proposed method? ... graph classification.
>
> **R6:** Thanks for your question. We would like to clarify that the proposed DeSGraDA does not assign an independent learnable weight to each edge in model training or inference, as prior work [3, 9]. In Eq.(1), $w_{ij}$ denotes the shared message-passing parameter between connected nodes, analogous to the aggregation weights used in conventional GNNs such as GCN or GIN. These weights are learned at the feature level and applied through the adjacency matrix $A$, rather than being edge-specific parameters. Therefore, the number of learnable parameters in DeSGraDA is comparable to that of standard GNNs and does not scale with the number of edges.  Moreover, since all graphs share the same learnable parameters and message-passing mechanism, the model can be trained on multiple graphs and naturally generalized to unseen graphs for classification tasks. Hence, DeSGraDA remains computationally efficient and fully applicable to graph classification scenarios.
>
> > W7: I am concerned about ... the GPU-based counterpart.
>
> **R7:** Thanks for your comment. Below, we provide details regarding the feasibility, estimation protocol, and comparison procedure.
>
> **1. Feasibility on Neuromorphic Hardware.**
> The proposed DeSGraDA is designed to be neuromorphic-compatible rather than directly deployed on the current ROLLS chip. The dynamic threshold and temporal attention mechanisms can be mapped to existing neuromorphic circuits that support event-driven membrane potential integration and adaptive firing thresholds. Specifically, the threshold modulation can be realized by adjusting local leak or reset parameters per neuron group, while the temporal attention corresponds to lightweight time-dependent integration of membrane potentials, as discussed in [4,5]. These processes require no dense matrix multiplications and operate within the asynchronous, event-driven computation paradigm of SNNs. Future work aims to deploy DeSGraDA on Loihi-2 or ROLLS-like chips for hardware validation.
>
> **2. Energy Estimation Protocol.**
> The reported energy consumption in Fig. 4a is derived using the standard estimation protocol established in prior SNN studies [3,4], where the total energy $E$ is approximated as: $E = N_\text{spike} \times E_\text{syn},$ where $N_{\text{spike}}$ denotes the total number of synaptic spike events during inference, and $E_{\text{syn}}$ represents the average energy per synaptic event measured from neuromorphic devices (e.g., ROLLS). This estimation method is widely used for evaluating neuromorphic efficiency without requiring full chip deployment. Although DeSGraDA’s architecture includes dynamic thresholds and temporal attention, both modules preserve sparse spike activity and event-driven computation, ensuring that the overall number of synaptic events remains comparable to conventional spiking networks.
>
> **3. GPU-based Counterpart Comparison.**
> For the GPU-based baselines, energy consumption was measured using the NVIDIA `nvidia-smi` profiler averaged over multiple inference runs. We followed the same procedure and evaluation metrics adopted in prior energy comparison works such as SpikingGCN [3] and related neuromorphic evaluations [4]. Thus, the GPU–SNN comparison in Section 5.4 is consistent with established benchmarking protocols. Even under this conservative estimation, DeSGraDA demonstrates significantly lower energy cost while achieving higher classification accuracy, underscoring the efficiency advantage of its event-driven design.

---

> ### Author Response · Authors · 2025-11-19
> **First round response of submission 17855**
>
> [1] Spiking transfer learning from rgb image to neuromorphic event stream. IEEE Transactions on Image Processing 2024.
>
> [2] pgesture: Source-free domain-adaptive semg-based gesture recognition with jaccard attentive spiking neural network. NIPS 2024.
>
> [3] Spiking graph convolutional networks. IJCAI 2022.
>
> [4] Neuromorphic architectures for spiking deep neural networks. IEDM 2015.
>
> [5] Spiking Neuron Models: Single Neurons, Plasticity 2002.
>
> [6] Spikformer: When spiking neural network meets transformer. ICLR 2023.
>
> [7] Spikingresformer: Bridging resnet and vision transformer in spiking neural networks. CVPR 2023.
>
> [8] Physics-informed attention temporal convolutional network for eeg-based motor imagery classification. IEEE Transactions on Industrial Informatics 2023.
>
> [9] Dynamic reactive spiking graph neural network. AAAI 2024.

---

> ### Author Response · Authors · 2025-11-27
>
> Dear Reviewer ZjWB,
>
> I hope this message finds you well. We noticed that the discussion phase is approaching its end, but we have not yet received any feedback regarding our rebuttal. We kindly ask if you could take a moment to review our responses at your earliest convenience.
>
> We sincerely hope that our clarifications address your concerns, and if there are any further questions or additional issues, we would be more than happy to provide further explanations.
>
> Thank you very much for your time and effort.
>
> Best regards,
>
> authors

---

### Official Review · Reviewer_sLij · 2025-11-02

**Soundness:** 3
**Presentation:** 3
**Contribution:** 3
**Rating:** 6
**Confidence:** 2

**Summary:**

This paper proposes DeSGraDA, a novel framework designed to address distribution shifts in Spiking Graph Networks (SGNs). DeSGraDA introduces a degree-conscious spiking representation with adaptive node thresholds, temporal distribution alignment via adversarial membrane potential matching, and pseudo-label distillation for reliable target supervision. The paper also provides a theoretical generalization bound for SGDA. Extensive experiments on multiple benchmark datasets demonstrate that DeSGraDA consistently surpasses state-of-the-art baselines in both classification accuracy and energy efficiency.

**Strengths:**

- This paper formally defines and proposes a novel framework for addressing SGDA, and introduces a biologically inspired degree-conscious mechanism that bridges the gap between spiking neural networks and graph learning.
- The proposed framework integrates three complementary modules: degree-conscious spiking representation, temporal distribution alignment, and pseudo-label distillation, supported by a derived generalization bound. This combination ensures both theoretical rigor and practical robustness.
- Extensive evaluations across multiple benchmark datasets demonstrate consistent improvements in both classification accuracy and energy efficiency. The results convincingly validate the model’s effectiveness and highlight its potential for low-power, real-world applications.

**Weaknesses:**

- The discussion of related work should include more recent DA methods or frameworks, such as [1,2], to better position this study within the current research landscape.
- In Section 4.3, the paper introduces a clustering-based approach for pseudo-label generation but does not specify which clustering algorithm is used or how DeSGraDA identifies the dominant pseudo-labels within each cluster.
- In Section 5.3, the ablation study appears incomplete, as it only evaluates the impact of removing individual components. It would be more comprehensive to also include results for removing two or three components simultaneously.

[1] Smoothness really matters: A simple yet effective approach for unsupervised graph domain adaptation. AAAI. 2025.
[2] Rethinking Graph Domain Adaptation: A Spectral Contrastive Perspective. ECML. 2025.

**Questions:**

- Can the proposed framework be applied to source-free domain adaptation scenarios?
- How does the presence of noise in the target domain affect the performance and robustness of the proposed framework?

---

> ### Author Response · Authors · 2025-11-19
> **First round response of submission 17855**
>
> We are truly grateful for the time you have taken to review our paper and your insightful review. Here we address your comments in the following:
>
> > W1: The discussion of related work should include more recent DA methods or frameworks, such as [1,2], to better position this study within the current research landscape.
>
> **R1:** Thanks for your suggestion. We include the recent GDA works [1,2] in the related work section to provide a more comprehensive view of the current landscape. While these studies focus on smoothness-based propagation and spectral contrastive alignment for conventional GNNs, our framework addresses spiking graph domain adaptation by modeling discrete spike dynamics and energy-efficient temporal alignment, which are not considered in [1,2].
>
> > W2: In Section 4.3, the paper introduces a clustering-based approach for pseudo-label generation but does not specify which clustering algorithm is used or how DeSGraDA identifies the dominant pseudo-labels within each cluster.
>
> **R2:** Thanks for your comment. In Section 4.3, DeSGraDA employs K-means clustering on the spiking-based target domain representations to generate pseudo-labels for unlabeled target samples. This clustering serves as a structural prior to guide pseudo-label refinement during adaptation. To determine the dominant label for each predicted class, we adopt a cluster-statistics-based strategy, where we analyze the distribution of cluster assignments within each predicted class and select the most frequent cluster label as the dominant one. This dominant label is then used to compute a class-consistent cross-entropy loss, enhancing the discriminability and alignment of target features.
>
> > W3: In Section 5.3, the ablation study appears incomplete, as it only evaluates the impact of removing individual components. It would be more comprehensive to also include results for removing two or three components simultaneously.
>
> **R3:** Thanks for your suggestion. We have conducted additional ablation experiments to evaluate the effects of removing multiple components simultaneously. Specifically, we examined four new variants: (1) DeSGraDA w CDAN and w/o PL, utilizing the static distribution alignment instead of temporal-based module and remove pseudo-label distillation modules; (2) DeSGraDA w CDAN and w/o TL, utilizing the static distribution alignment instead of temporal-based module while using fixed global thresholds; (3) DeSGraDA w/o PL and TL, removing pseudo-label distillation and using fixed thresholds; and (4) DeSGraDA w CDAN, w/o PL, and TL, removing all three modules and retaining only the backbone with fixed thresholds.
>
> | **Methods**                    | **P0→P1** | **P1→P0** | **P0→P2** | **P2→P0** | **P0→P3** | **P3→P0** | **P1→P2** | **P2→P1** | **P1→P3** | **P3→P1** | **P2→P3** | **P3→P2** |
> | ------------------------------ | --------- | --------- | --------- | --------- | --------- | --------- | --------- | --------- | --------- | --------- | --------- | --------- |
> | DeSGraDA w CDAN and w/o PL     | 72.8      | 81.0      | 66.3      | 76.3      | 72.7      | 77.6      | 65.8      | 69.6      | 71.0      | 68.8      | 73.6      | 67.8      |
> | DeSGraDA w CDAN and w/o TL     | 73.1      | 80.3      | 65.2      | 75.9      | 72.5      | 78.0      | 63.9      | 69.0      | 69.5      | 68.2      | 73.3      | 64.9      |
> | DeSGraDA w/o PL and TL         | 72.2      | 79.9      | 65.4      | 78.6      | 73.1      | 77.7      | 63.0      | 69.9      | 69.0      | 68.0      | 72.4      | 63.7      |
> | DeSGraDA w CDAN, w/o PL and TL | 71.4      | 78.7      | 63.5      | 74.2      | 71.2      | 76.2      | 62.3      | 68.0      | 68.4      | 67.3      | 70.5      | 63.0      |
> | **DeSGraDA**                   | **76.3**  | **84.6**  | **69.2**  | **83.6**  | **77.5**  | **83.7**  | **69.8**  | **74.0**  | **76.2**  | **73.0**  | **77.8**  | **70.5**  |
>
> The results clearly show that removing multiple components leads to a consistent decline in performance, indicating that each part of DeSGraDA plays an essential role in its overall effectiveness. When all three components are removed, the performance drops sharply, confirming that the improvement stems from their combined contributions rather than any single component. We have revised our manuscript in Section 5.3.

---

> > ### Author Response · Authors · 2025-11-19
> > **First round response of submission 17855**
> >
> > > Q1: Can the proposed framework be applied to source-free domain adaptation scenarios?
> >
> > **R4:** Thanks for your question. The current framework is developed for the graph domain adaptation (GDA) setting, where both source and target features are utilized during training to perform temporal distribution alignment and pseudo-label refinement. Therefore, the present version requires access to source-domain data. In future work, we plan to extend DeSGraDA to the source-free domain adaptation scenario by adapting the degree-conscious and temporal alignment mechanisms to operate without direct access to source features.
> >
> > >Q2: How does the presence of noise in the target domain affect the performance and robustness of the proposed framework?
> >
> > **R5:** Thanks for your question. The presence of noise in the target domain does not affect the proposed DeGraDA, as DeSGraDA operates under an unsupervised domain adaptation setting where no target-domain labels are used during training. Since pseudo-labels are generated based on model consistency rather than ground-truth labels, label noise in the target domain does not directly influence the adaptation process or degrade model robustness.
> >
> > -----
> >
> > [1] Smoothness really matters: A simple yet effective approach for unsupervised graph domain adaptation. AAAI 2025.
> >
> > [2] Rethinking Graph Domain Adaptation: A Spectral Contrastive Perspective. ECML 2025.

---

> ### Author Response · Authors · 2025-11-27
>
> Dear Reviewer sLij,
>
> I hope this message finds you well. We noticed that the discussion phase is approaching its end, but we have not yet received any feedback regarding our rebuttal. We kindly ask if you could take a moment to review our responses at your earliest convenience.
>
> We sincerely hope that our clarifications address your concerns, and if there are any further questions or additional issues, we would be more than happy to provide further explanations.
>
> Thank you very much for your time and effort.
>
> Best regards,
>
> authors

---

> > ### Comment · Reviewer_sLij · 2025-11-27
> >
> > I really appreciate the author's excellent and detailed responses, which have convincingly addressed all my concerns and demonstrate their effectiveness and robustness. Therefore, I have accordingly increased my score.

---

> > > ### Author Response · Authors · 2025-11-27
> > >
> > > Thank you for your positive feedback and for the time and effort you dedicated to reviewing our paper. We appreciate your constructive comments and are glad our responses addressed your concerns.

---

### Author Response · Authors · 2025-12-04
**Rebuttal Summary for AC**

Dear Area Chair,

We sincerely appreciate the time and effort devoted to reviewing our submission.

Below is a concise summary of the reviewers’ positive assessments, followed by how we addressed their concerns and clarified the issue regarding the mismatched review from 5EVp.

## Key Strengths Highlighted by Reviewers

**1. Novel problem formulation and well motivated framework.**

 Reviewers sLij and TxSC both recognized that the paper introduces the spiking graph domain adaptation setting in a clear and well motivated manner. They appreciated that the work identifies fundamental limitations of existing SGNs under distribution shift and proposes a principled framework to address them.

**2. Effective integration of complementary components.**

Reviewer sLij highlighted that the combination of degree conscious spiking representation, temporal distribution alignment, and pseudo label distillation forms a coherent and well justified architecture. Reviewer TxSC noted that the proposed modules complement each other and contribute to the final performance in a meaningful way.

**3. Strong empirical performance across diverse domain shifts.**

Reviewer sLij emphasized that DeSGraDA achieves consistent gains across both structural and feature based domain shifts, showing clear advantages over a wide range of baselines. Reviewer ZjWB also acknowledged that the experiments demonstrate meaningful improvements compared with multiple competing approaches. Reviewer TxSC further commented that the evaluations are thorough and convincing across datasets.

**4. Clear motivation and evidence for degree aware representations.**

Reviewers sLij and TxSC both recognized that the paper provides solid motivation for incorporating node degree into spiking threshold adaptation. They pointed out that the analysis of bias introduced by fixed thresholds and the accompanying empirical evidence make the rationale of the degree conscious design convincing.

## Reviewers’ Concerns We Addressed

**1. Completeness of related work**

Reviewer sLij suggested incorporating more recent GDA studies to better contextualize our contribution.

We expanded the related work section to include recent smoothness based and spectral contrastive GDA methods and clarified how our spiking based SGDA setting fundamentally differs from existing GNN based approaches.

**2. Clarification of the pseudo label clustering process**

Reviewer sLij requested more details about the clustering algorithm and dominant label selection.

We updated Section 4.3 to specify the use of K means on spiking representations and a cluster statistics based strategy for identifying dominant labels, ensuring the procedure is fully transparent.

**3. More comprehensive ablation studies**

Reviewer sLij asked for ablations removing multiple components simultaneously.

We added four new multi component ablations, all showing consistent performance degradation and confirming the necessity of each module in DeSGraDA.

**4. Practicality and implementation assumptions**

Reviewer ZjWB raised concerns regarding platform assumptions, feasibility of adaptive thresholds, and the use of attention.

We clarified that DeSGraDA is implemented within a standard GPU based SGN setting, that thresholds operate at the degree level rather than per neuron, and that temporal attention is used only at the graph level readout, preserving the spiking computation flow.

**5. Architectural clarity and fairness of comparison**

Reviewer ZjWB requested clarification about parameterization and backbone fairness.

We revised the methodology section to clearly explain the model structure and ensured all baselines share comparable architectures, preventing structural advantages.

## Mismatched Review from Reviewer 5EVp

Reviewer 5EVp initially submitted a review that was clearly misassigned and referred to a different paper rather than our submission.

## Summary

The submission clearly introduces the SGDA problem, proposes a biologically inspired degree conscious mechanism, provides theoretical justification, and validates the approach through extensive experiments. The concerns raised primarily centered on related work coverage, clarity of the pseudo label procedure, completeness of ablation studies, and several implementation assumptions. All of these points have been thoroughly addressed through targeted revisions and expanded explanations, which further strengthen the work.

During the discussion phase, **reviewer sLij explicitly stated that their concerns had been fully resolved and subsequently increased their score from 6 to 8**. Reviewer ZjWB maintained their original assessment after reviewing the rebuttal. Reviewer TxSC did not provide additional comments during the discussion window. We thank all reviewers and the AC once again for their constructive feedback, which has significantly strengthened our submission.

Best wishes,

All authors.

---

### Meta-Review · Area_Chair_dykZ · 2026-01-07

**Summary:**

This submission introduces spiking graph domain adaptation (SGDA) and proposes DeSGraDA, combining (i) degree-conscious thresholds, (ii) temporal adversarial alignment in membrane-potential space, and (iii) pseudo-label distillation, plus a generalization bound. Reviews agreed the paper is clearly written and empirically strong on the provided benchmarks, but the final decision hinges on whether the contribution is broad and well-grounded enough for ICLR, especially given serious concerns about scope, practicality, and the evidence supporting the energy/neuromorphic claims.

**Reviewer Concerns:**

**Concerns addressed in the rebuttal/discussion:**

1. Pseudo-labeling details and reliability: clarified clustering (K-means), dominant-label selection, and filtering logic; TxSC’s questions were answered and they acknowledged the clarifications.
2. Ablations: added multi-component removal ablations showing consistent drops, supporting that gains come from the combination rather than a single trick.
3. Positioning vs prior work (SpikeNet adaptive thresholds): clarified that their mechanism is degree/structure-based rather than global temporal thresholding; TxSC explicitly said this distinction is now clear.
4. Related work coverage and terminology: expanded DA references; fixed inconsistent terms.

**Remaining concerns:**

1. Scope and impact: the intersection “spiking + graph + domain adaptation” remains narrow, and the practical case is not fully convincing to all reviewers (ZjWB).
2. Neuromorphic feasibility and energy claims: the work argues neuromorphic compatibility, but the most skeptical reviewer remained unconvinced about how the full pipeline (dynamic thresholds, temporal attention, alignment) maps to real neuromorphic constraints, and the energy comparison methodology (GPU measurements vs event-based estimates) is still not airtight or directly validated on hardware (ZjWB).
3. Fairness/clarity of comparisons: despite the authors’ claim of matched backbones/protocols, some architectural/implementation ambiguity remains for the most critical reviewer (ZjWB).

**Important point:**  Review process noise -- one review (5EVp) was clearly misassigned and should be disregarded; this reduces signal quality by this reviewer.

**Reviewer Scores:**

sLij: increased to 8 after rebuttal.

TxSC: remained 6, acknowledged concerns addressed, but still viewed novelty/contribution as only marginally above threshold.

ZjWB: remained 2 core concerns about scope, platform realism, and energy/feasibility not resolved to their satisfaction.

5EVp: invalid/mismatched review; although the reviewer update their review, I think they should not be fully counted (and hence no meaningful score update possible). Their score is 2.

Despite a strong positive reviewer and a solid second, the remaining negative review raises foundational issues (scope and especially the strength/credibility of the neuromorphic and energy-efficiency claims) that are not fully settled by the rebuttal. The paper may be promising, but the current evidence does not justify acceptance at ICLR given the uncertainty around the claimed deployment story, in my opinion.

I lean towards rejection based on the unresolved feasibility/energy and scope concerns, but acknowledge that with two supportive reviews (one strongly supportive), an acceptance is defensible if the venue prioritizes novelty in this niche.

---

### Decision · Program_Chairs · 2026-01-26

Reject